# Trajectory Generation with Conservative Value Guidance for Offline Reinforcement Learning

**Tieru Wang**     **Kunbao Wu**     **Guoshun Nan**[*]

Beijing University of Posts and Telecommunications, China

{wangtieru,wukunbao,nanguo2021}@bupt.edu.cn

## Abstract

Recent advances in offline reinforcement learning (RL) have led to the development of high-performing algorithms that achieve impressive results across standard benchmarks. However, many of these methods depend on increasingly complex planning architectures, which hinder their deployment in real-world settings due to high inference costs. To overcome this limitation, recent research has explored data augmentation techniques that offload computation from online decision-making to offline data preparation. Among these, diffusion-based generative models have shown potential in synthesizing diverse trajectories but incur significant overhead in training and data generation. In this work, we propose Trajectory Generation with Conservative Value Guidance (TGCVG), a novel trajectory-level data augmentation framework that integrates a high-performing offline policy with a learned dynamics model. To ensure that the synthesized trajectories are both high-quality and close to the original dataset distribution, we introduce a value-guided regularization during the training of the offline policy. This regularization encourages conservative action selection, effectively mitigating distributional shift during trajectory synthesis. Empirical results on standard benchmarks demonstrate that TGCVG not only improves the performance of state-of-the-art offline RL algorithms but also significantly reduces training and trajectory synthesis time. These findings highlight the effectiveness of value-aware data generation in improving both efficiency and policy performance. Our code is available at https://github.com/wangtieru2/TGCVG.

## 1 Introduction

Offline Reinforcement Learning (RL) (Kumar et al., 2020; Fujimoto & Gu, 2021; Kostrikov et al., 2021b), which focuses on training a policy from the static dataset, is emerging as the critical solution for environments where online interaction is costly or risky. The main challenge of offline RL is the distributional shift which would cause extrapolation error (Fujimoto et al., 2019). To mitigate this problem, many previous works add the constraints to estimated value functions (Kumar et al., 2020; Lyu et al., 2022) and policy networks (Wang et al., 2020; Fujimoto & Gu, 2021), or learn without querying out-of-distribution (OOD) samples (Kostrikov et al., 2021b; Hansen-Estruch et al., 2023). Moreover, some works have introduced strong generative architectures like Diffusion and Transformer (Wang et al., 2022; Chen et al., 2024; Kim et al., 2023; Hu et al., 2024) for richer representation learning to train the policy.

However, these methods often involve increased model and algorithmic complexity, leading to longer inference time and raising concerns about their practicality in real-world deployment. To address this issue, recent studies have explored data augmentation techniques that aim to broaden the dataset distribution and improve the performance of simpler offline RL algorithms (Kumar et al., 2020; Fujimoto & Gu, 2021; Kostrikov et al., 2021b). Early works primarily enrich the data distribution by adding noise to states (Laskin et al., 2020; Sinha et al., 2022), but such perturbations are limited in diversity due to their constrained range. More recent approaches introduce generative models, particularly Diffusion models (Lu et al., 2023; Li et al., 2024; Lee et al., 2024; Yang &

---

[*]Corresponding author.

Wang, 2025), which offer a stronger mechanism for synthesizing diverse and dynamics-consistent samples. However, Diffusion models are computationally expensive to train and require multi-step denoising for accurate generation. Moreover, they typically lack explicit guidance toward high-value regions, resulting in limited improvements in data quality and marginal gains in offline policy performance.

To address these challenges, we propose a simple and effective data synthesizer for offline RL, termed Trajectory Generation with Conservative Value Guidance (TGCVG). Our method uses a Transformer-based policy network trained with Conservative Q-learning (CQL) to generate high-quality actions. A dynamics model from model-based RL (Lin et al., 2024) is then employed to predict the corresponding rewards and next states, yielding a set of synthetic transitions. Intuitively, this process mimics the original dataset collection procedure, where an online policy interacts with the environment to generate trajectories (Fu et al., 2020; Hafner et al., 2020). Similar to prior works (Li et al., 2024; Lee et al., 2024), our method generates sequential data, but replaces the Diffusion module with the Transformer to significantly reduce computational overhead. Transformer has been shown to be effective in offline RL tasks (Chen et al., 2021; Hu et al., 2024; Kim et al., 2024), making them a practical and efficient alternative. The key component of TGCVG is conservative value guidance, which constrains each generated $(s_t, a_t, s_{t+1})$ tuple to lie within the dataset distribution. This limits OOD risk to a single step at each model interaction and prevents its accumulation across rollouts. As a result, TGCVG produces dynamically consistent trajectories that are more stable and reliable for offline policy training.

Our Contributions are summarized as follows. (1) We propose TGCVG, a novel value-guided data synthesizer that generates high-quality trajectories while significantly reducing computational overhead. (2) We demonstrate that TGCVG consistently achieves state-of-the-art performance across a wide range of offline RL benchmarks, validating the effectiveness of our method. (3) We provide a comprehensive ablation study to analyze the impact of key design choices and demonstrate the robustness of our pipeline.

## 2 RELATED WORK

### 2.1 MODEL-FREE OFFLINE REINFORCEMENT LEARNING

Model-free offline RL algorithms aim to train policy networks directly from static datasets. Several methods (Fujimoto & Gu, 2021; Kostrikov et al., 2021b; Lyu et al., 2022) adopt the value-based paradigm from online RL by learning Q-functions to guide behavior improvement. To address distributional shift, value-based approaches typically incorporate explicit policy constraints (Kumar et al., 2019; Wu et al., 2019; Fujimoto & Gu, 2021), penalize overestimated Q-values on OOD samples (Kumar et al., 2020; Kostrikov et al., 2021a; Wu et al., 2021), or avoid querying OOD actions (Wang et al., 2018; Kostrikov et al., 2021b; Hansen-Estruch et al., 2023). Beyond these methods, generative models such as Transformer (Chen et al., 2021; Kim et al., 2023) and Diffusion models (Janner et al., 2022; Ajay et al., 2022) have been introduced for modeling offline RL tasks. Recent works further explore hybrid approaches that combine generative modeling with value-based techniques (Wang et al., 2022; 2024; Hu et al., 2024; Gao et al., 2025) to enhance policy improvement. However, as algorithmic architectures grow increasingly complex, the computational cost of policy evaluation rises substantially, limiting the practical deployment of offline RL algorithms.

### 2.2 MODEL-BASED OFFLINE REINFORCEMENT LEARNING

Model-based offline RL algorithms aim to learn a dynamics model from static datasets and leverage it to derive or improve policies. A central challenge in this setting is how to effectively utilize the learned model. Prior works address this by quantifying model uncertainty (Yu et al., 2020; Sun et al., 2023), learning conservative value functions (Yu et al., 2021; Jeong et al., 2022), or transforming one-step models into multi-step transition models (Machado et al., 2023; Lin et al., 2024). While data augmentation methods (Lu et al., 2023; Lee et al., 2024) also generate synthetic trajectories, they decouple data generation from policy learning. This modularity enables the synthesized data to be reused by model-free algorithms, offering greater flexibility and generalization. Notably, such methods demonstrate superior scalability in high-dimensional environments, where model-based approaches often suffer from compounding model errors and poor generalization.

## 2.3 DATA AUGMENTATION IN OFFLINE REINFORCEMENT LEARNING

Data augmentation has emerged as an important technique for improving the performance of offline RL algorithms. Traditional approaches (Yarats et al., 2021; Laskin et al., 2020; Sinha et al., 2022) apply simple transformations such as noise injection or random translations to pixel-based or state-based observations. With the rise of generative models, recent works have explored generative data augmentation. Several studies (Lu et al., 2023; He et al., 2023; Li et al., 2024; Yang & Wang, 2025) leverage Diffusion models to approximate the data distribution and synthesize dynamically plausible transitions. Building on Diffusion-based data augmentation, GTA (Lee et al., 2024) further proposes generating high-return trajectories to improve overall dataset quality. This reward-guided design has been shown to enhance the performance of offline RL algorithms. Our TGCVG differs from prior methods in two aspects: (1) it replaces costly Diffusion models with a lightweight Transformer-based policy for efficient sequential data generation; (2) it leverages conservative value guidance to steer synthesis toward high-return trajectories while remaining within the dataset distribution.

## 3 PRELIMINARIES

### 3.1 OFFLINE REINFORCEMENT LEARNING

Reinforcement learning is typically formalized as a Markov Decision Process (MDP) (Bellman, 1957), defined by the tuple $(\rho_0, \mathcal{S}, \mathcal{A}, P, R, \gamma)$, where $\rho_0$ denotes the initial state distribution, $\mathcal{S}$ and $\mathcal{A}$ are the state and action spaces, $P(s'|s, a)$ is the transition probability, $R(s, a)$ is the reward function, and $\gamma$ is the discount factor. The objective of standard RL is to learn a policy $\pi^*(a|s)$ that maximizes the expected return $\mathbb{E}[\sum_{t=0}^{\infty} \gamma^t r(s_t, a_t)]$ through direct interaction with the environment. In contrast, offline RL focuses on learning from a fixed dataset $\mathcal{D} = \{(s, a, r, s')\}$ collected by a behavior policy $\pi_\beta$, without further environment interaction. This setting poses unique challenges, as the agent must generalize from static data without the ability to explore.

### 3.2 TRANSFORMER IN OFFLINE REINFORCEMENT LEARNING

Transformer-based methods typically frame offline RL as a supervised learning problem to improve training stability. Among them, Decision Transformer (DT) (Chen et al., 2021) is a seminal approach that models trajectories as sequences of states, actions, and returns-to-go (RTGs), where the RTG at timestep $t$ is defined as the sum of future rewards: $\hat{R}_t = \sum_{t'=t}^{T} r_{t'}$. At each timestep $t$, DT takes as input a sequence $\tau = (\hat{R}_{t-L+1}, s_{t-L+1}, a_{t-L+1}, \cdots, \hat{R}_{t-1}, s_{t-1}, a_{t-1}, \hat{R}_t, s_t)$, where $L$ denotes the sequence length, and predicts the next action $a_t$. During training, the model is optimized to match the ground-truth actions from the dataset. In the evaluation stage, since true RTGs are unavailable, DT conditions on a pre-specified target RTG that represents the desired return. At each step, the received reward is subtracted from the target RTG until the episode ends or the maximum trajectory length is reached.

While Transformer-based methods benefit from leveraging long-term historical information for decision-making, their lack of explicit value guidance can hinder effective trajectory stitching. To overcome this limitation, recent work has explored integrating value-based learning with sequence modeling to improve policy optimization (Yamagata et al., 2023; Gao et al., 2024; Hu et al., 2024; Kim et al., 2024), effectively improving decision quality and leading to stronger empirical performance.

## 4 METHOD

In this section, we propose Trajectory Generation with Conservative Value Guidance (TGCVG), a Transformer-based generative framework that interacts with a pretrained dynamics model to produce high-quality offline trajectories. We first describe how to train the Transformer policy using conservative Q-value guidance. Then, we introduce the pipeline for trajectory-level data generation using the learned policy and the dynamics model. The overall architecture is summarized in Figure 1. Finally, we employ standard offline RL algorithms on the augmented dataset to improve performance.

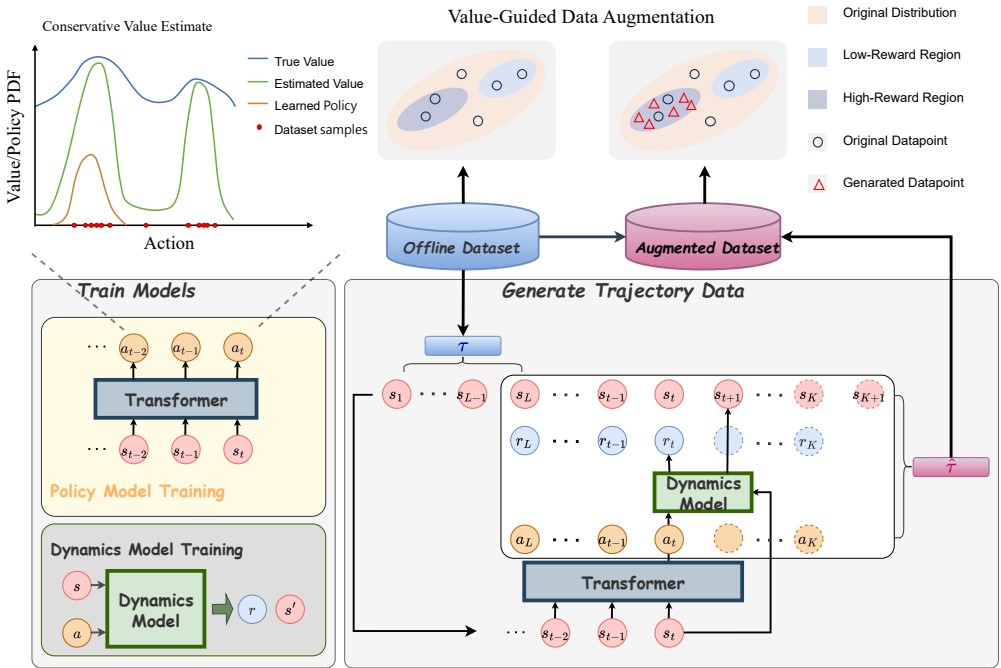

Figure 1: Overview of the TGCVG Architecture. In the first stage, we train a conservative policy network and a dynamics model to facilitate data generation. In the subsequent stage, we exploit the interaction between these components to synthesize new trajectories that align with the high-reward distribution of the offline dataset.

## 4.1 CONSERVATIVE Q-LEARNING BASED ON TRANSFORMER

Building upon Transformer-based architectures, we explore the integration of return-conditioned supervised learning (RCSL) (Brandfonbrener et al., 2022) with Q-learning to optimize the policy network. To streamline training, we eliminate action and RTG tokens according to DD (Ajay et al., 2022) and our ablation study in Appendix B.1, which demonstrates that RTGs are unnecessary when employing Q-value regularization. The resulting trajectory representation, consisting solely of states, is defined as:

$$\tau_t = (s_{t-L+1}, \cdots, s_{t-1}, s_t), \tag{1}$$

where $L$ represents the trajectory length. Adopting the conservative value estimation technique from CQL (Kumar et al., 2020), we employ an ensemble of five networks: two Q-networks $Q_{\phi_1}, Q_{\phi_2}$, two target networks $Q_{\phi'_1}, Q_{\phi'_2}$ and the policy network $\pi_\theta$. The Q-networks are trained by solving the following optimization problem:

$$\min_\phi \lambda \mathbb{E}_{s_i \sim \mathcal{D}, a_i \sim \mu(\cdot|s_i)} \left[ \log \sum_{a_i} \exp(Q_{\phi_i}(s_i, a_i)) - \mathbb{E}_{a_i \sim \hat{\pi}_\beta(\cdot|s_i)}[Q_{\phi_i}(s_i, a_i)] \right]$$

$$+ \frac{1}{2} \mathbb{E}_{(\tau_t, \tau_{t+1}, a_{t-L+1:t}, r_{t-L+1:t}) \sim \mathcal{D}} \sum_{m=t-L+1}^{t} \left\| \hat{Q}_m - Q_{\phi_i}(s_m, a_m) \right\|^2, \tag{2}$$

$$\text{where } \hat{Q}_m = \sum_{j=m}^{t} \gamma^{j-m} r_j + \gamma^{t+1-m} \min_{i=1,2} Q_{\phi'_i}(s_{t+1}, \hat{a}_{t+1}).$$

In this formulation, $\mu(\cdot|s_i)$ is used to match the marginal distribution in the dataset, $\hat{a}_{t+1}$ is sampled from the policy $\pi_\theta$, $\hat{\pi}_\beta(\cdot|s_i)$ denotes the behavior policy, $\gamma$ is the discount factor, and $\lambda$ is a weighting coefficient. We adopt the n-step Bellman backup to estimate the Q-value function, which has been shown to outperform the 1-step approximation in recent studies (Hu et al., 2024). The first term in

---

**Algorithm 1** Transformer-CQL

---

**Input:** Sequence horizon $L$, offline datasets $\mathcal{D}$, coefficient $\rho$.
Initialize policy network $\pi_\theta$, critic networks $Q_{\phi_1}$ and $Q_{\phi_2}$, and target networks $Q_{\phi'_1}$ and $Q_{\phi'_2}$.
**for** $t = 1$ **to** $T$ **do**
    Sample mini-batch $\mathcal{B} = \{(s_j, a_j, r_j)_{j=t}^{t+L}\} \sim \mathcal{D}$.
    // Q-value function learning
    Sample $\hat{a}_{t+L} \sim \pi_\theta(\hat{a}_{t+L}|s_{t+1:t+L})$.
    Update $Q_{\phi_1}$ and $Q_{\phi_2}$ by Equation 2.
    // Policy learning
    **for** $i = 0$ **to** $L - 1$ **do**
        Sample $\hat{a}_{t+i} \sim \pi_\theta(\hat{a}_{t+i}|s_{t:t+i})$.
    **end for**.
    Update policy by minimizing Equation 3.
    // Update target networks
    $\phi'_i = \rho\phi'_i + (1 - \rho)\phi_i$ for $i = \{1, 2\}$.
**end for**.

---

the objective serves as a regularization component, penalizing Q-values for OOD state-action pairs while preserving values for those within the dataset distribution.

The policy learning objective is defined as:

$$\mathcal{L}(\theta) = \min_\theta \mathbb{E}_{\tau_t \sim \mathcal{D}} \mathbb{E}_{s_i \sim \tau_t, \hat{a}_i \sim \pi_\theta(\cdot|\tau_t)_i} \Big[ \alpha \log \pi_\theta(\hat{a}_i|s_i) - \min_{i=1,2} Q_{\phi_i}(s_i, \hat{a}_i) \Big],$$

$$\text{where } \alpha = \arg\min_\alpha \mathbb{E}_{\tau_t \sim \mathcal{D}} \mathbb{E}_{s_i \sim \tau_t, \hat{a}_i \sim \pi_\theta(\cdot|\tau_t)_i} \Big[ -\log \alpha \cdot (\log \pi_\theta(\hat{a}_i|s_i) + \mathcal{H}_{\text{target}}) \Big], \tag{3}$$

where $\mathcal{H}_{\text{target}}$ is predetermined by the environment's action space. The policy loss follows the SAC (Haarnoja et al., 2018) framework to promote policy improvement while encouraging sufficient exploration. We detail the learning procedure in Algorithm 1. In addition to our conservative value-based optimization, an alternative Transformer-based Q-learning approach exists, which incorporates value improvement into the supervised learning objective by adopting the policy constraint paradigm from TD3BC (Hu et al., 2024). For clarity, we refer to these two approaches as Transformer-CQL and Transformer-TD3BC, respectively. In this work, we ultimately adopt Transformer-CQL, and the rationale for this choice is discussed in Section 5.3.

## 4.2 GENERATING SEQUENCE DATA WITH THE DYNAMICS MODEL

In the data generation phase, we mimic the online data collection process, where an agent interacts with the environment to collect $(s, a, r, s')$ tuples. Specifically, we use the learned policy network to interact with a pretrained dynamics model (Lin et al., 2024). To generate synthetic transitions, we first sample a state sequence of length $K$ from the original dataset:

$$\tilde{\tau} = (s_{t-K+1}, \cdots, s_{t-1}, s_t). \tag{4}$$

We then take the first $L$ states from this sequence, denoted as $\tau_{t-K+L} = (s_{t-K+1}, \cdots, s_{t-K+L})$, and feed them into the policy network to obtain the action $\hat{a}_{t-K+L}$. The next state and reward are predicted by the dynamics model $f_\omega$ as:

$$\hat{s}_{t-K+L+1}, \hat{r}_{t-K+L} = f_\omega(s_{t-K+L}, \hat{a}_{t-K+L}), \tag{5}$$

The predicted state $\hat{s}_{t-K+L+1}$ is appended to the state sequence, forming a new window $\tau_{t-K+L+1} = (s_{t-K+2}, \cdots, \hat{s}_{t-K+L+1})$, which is used as input to the Transformer-based policy. This autoregressive process is repeated until we construct the full generated trajectory:

$$\hat{\boldsymbol{\tau}} = \begin{bmatrix} s_{t-K+L}, & \cdots, & \hat{s}_{t-1}, & \hat{s}_t, & \hat{s}_{t+1} \\ \hat{a}_{t-K+L}, & \cdots, & \hat{a}_{t-1}, & \hat{a}_t, & 0 \\ \hat{r}_{t-K+L}, & \cdots, & \hat{r}_{t-1}, & \hat{r}_t, & 0 \end{bmatrix}. \tag{6}$$

Other auxiliary information (e.g., terminal indicators) is directly aligned with and inherited from the corresponding timesteps in the original sequence, thereby preserving the original episode termination signals. The generated trajectories are fused with original trajectories, leading to our augmented dataset.

### 4.3 TRAINING OFFLINE RL ALGORITHMS WITH AUGMENTED DATASET

Following the data generation phase, we obtain a set of new, high-quality trajectories. Consistent with prior works (Lu et al., 2023; Lee et al., 2024), we combine the generated trajectories with the original dataset via random shuffling. The resulting augmented dataset is then used to train standard offline RL algorithms.

## 5 EXPERIMENTS

In this section, we present a series of experiments to evaluate the effectiveness of TGCVG. We begin by outlining the experimental setup and demonstrate that our method consistently enhances the performance of offline RL algorithms across a range of environments. We then perform ablation studies to assess the contribution of each component and provide a rationale for our design choices. Finally, we investigate key characteristics of TGCVG to gain deeper insights into its behavior.

### 5.1 EXPERIMENTAL SETUP

**Datasets.** We conduct experiments on the D4RL benchmark (Fu et al., 2020), including MuJoCo Gym, Maze2D, and AntMaze tasks. The MuJoCo locomotion suite consists of widely used benchmark tasks in offline RL, characterized by smooth reward functions and a high proportion of near-optimal trajectories. The Maze2D datasets are designed to evaluate an algorithm's ability to effectively stitch sub-trajectories in navigation tasks. AntMaze presents a more challenging maze navigation environment, featuring sparse 0–1 rewards and higher-dimensional state and action spaces compared to Maze2D.

**Baselines.** We compare our method against several data augmentation baselines: (1) S4RL (Sinha et al., 2022), which perturbs states with Gaussian noise; (2) Synther (Lu et al., 2023), which employs a diffusion model to learn the transition distribution from offline data and generate new samples; (3) GTA (Lee et al., 2024), which augments offline data via a partial noising and denoising process guided by amplified return signals. The performance scores for these baseline methods are mainly sourced from results published in (Lee et al., 2024), ensuring a fair comparison.

**Offline RL Algorithms.** We evaluate our TGCVG using several representative algorithms: (1) CQL (Kumar et al., 2020), which penalizes Q-values on unseen actions to enforce conservatism; (2) TD3BC (Fujimoto & Gu, 2021), which constrains the learned policy to stay close to the behavior policy; (3) IQL (Kostrikov et al., 2021b), which performs implicit value regularization by querying only in-distribution actions; and (4) DT (Chen et al., 2021), which takes trajectory data as input and applies Transformer to model the distribution of trajectories in the offline dataset. Following GTA, we employ only IQL for the Maze2D and AntMaze datasets, as it provides complete hyperparameter settings and demonstrates stable performance in these environments.

### 5.2 MAIN RESULTS

Tables 1 and 2 report the normalized scores across three task domains, following the protocol of (Fu et al., 2020). We analyze results by domain.

**Gym Domain.** Our TGCVG consistently boosts the performance of all offline RL algorithms, with CQL and TD3BC achieving results comparable to recent state-of-the-art methods (Hu et al., 2024; Kim et al., 2024). This highlights the effectiveness of the policy used for data synthesis and demonstrates that the conservative value guidance effectively mitigates distributional shift. Gains are especially notable on suboptimal datasets (e.g., *medium*, *medium-replay*), where trajectory stitching is more critical, indicating the advantage of our Transformer-CQL design in imperfect data regimes.

**Maze2D and AntMaze Domains.** In Maze2D, which evaluates trajectory stitching capabilities, our TGCVG generates useful data across both simple and complex environments. In the more challenging AntMaze tasks, which are characterized by sparse rewards and high-dimensional state spaces, TGCVG still enables model-free learners to perform competitively, even though it employs similar dynamics models as those used in previous model-based approaches (Yu et al., 2020; Sun et al., 2023; Lin et al., 2024). These results suggest that placing greater emphasis on policy learning may be more beneficial than solely refining dynamics models.

Table 1: Normalized results of TGCVG and baselines on the MuJoCo datasets. Average and standard deviation scores are reported over 5 seeds and the best average values are marked in bold. The dataset names are abbreviated as follows: *medium* to 'm', *medium-replay* to 'm-r', and *medium-expert* to 'm-e'.

| Algo. | Aug. | halfcheetah | | | hopper | | | walker2d | | | Average |
|---|---|---|---|---|---|---|---|---|---|---|---|
| | | m | m-r | m-e | m | m-r | m-e | m | m-r | m-e | |
| TD3BC | None | 48.42±0.62 | 44.64±0.71 | 89.48±5.50 | 61.04±3.18 | 65.69±24.41 | 104.08±5.81 | 84.58±1.92 | 84.11±4.12 | 110.23±0.37 | 76.92 |
| | S4RL | 48.74±0.31 | 44.53±0.30 | 90.78±4.65 | 59.34±3.50 | 67.39±23.81 | 106.10±7.24 | 84.63±2.44 | 83.42±4.70 | 110.21±0.35 | 77.24 |
| | SynthER | 49.16±0.39 | 45.57±0.34 | 85.47±11.35 | 63.70±3.69 | 78.81±15.80 | 98.99±11.27 | 85.43±1.14 | 90.67±1.56 | 109.95±0.32 | 78.64 |
| | GTA | 57.84±0.51 | 50.04±0.84 | 93.13±3.07 | 69.57±4.05 | 89.31±16.84 | **110.40**±4.04 | 86.69±0.89 | 93.82±1.74 | **110.86**±0.34 | 84.63 |
| | TGCVG | **68.14**±0.58 | **57.78**±0.32 | **94.68**±0.67 | **90.47**±4.06 | **94.08**±15.35 | 106.73±6.74 | **87.50**±0.20 | **94.23**±0.73 | 110.12±0.52 | **89.30** |
| CQL | None | 46.98±0.20 | 44.70±0.51 | 95.90±0.52 | 61.13±3.20 | 82.33±16.37 | 104.38±7.30 | 82.26±1.18 | 79.74±5.19 | 109.50±0.43 | 78.55 |
| | S4RL | 47.00±0.23 | 44.62±0.42 | 95.89±0.45 | 62.72±3.46 | 78.82±9.89 | 108.87±2.69 | 81.46±2.28 | 80.82±8.35 | 109.65±0.22 | 78.87 |
| | SynthER | 47.21±0.14 | 46.03±0.40 | 95.29±1.90 | 64.65±4.78 | 92.06±13.40 | 107.66±6.68 | 81.91±0.89 | 86.62±3.03 | 109.36±0.36 | 81.20 |
| | GTA | 54.14±0.31 | 51.36±0.27 | 94.93±3.71 | 74.80±7.42 | 98.88±3.51 | 110.90±3.44 | 80.40±4.98 | 91.57±5.15 | **110.44**±0.28 | 85.27 |
| | TGCVG | **68.31**±0.27 | **59.03**±0.28 | **97.40**±0.47 | **88.25**±2.99 | **101.80**±0.13 | **111.36**±1.39 | **85.69**±0.34 | **92.91**±1.22 | 109.79±0.34 | **90.50** |
| IQL | None | 48.65±0.19 | 43.35±0.50 | 94.57±1.88 | 66.35±7.09 | 95.76±4.01 | 91.69±25.97 | 84.34±3.31 | 69.60±10.80 | 112.37±0.60 | 78.52 |
| | S4RL | 48.58±0.29 | 43.57±0.65 | 94.22±1.59 | 65.06±5.94 | 86.72±22.01 | 99.82±8.09 | **84.58**±4.26 | 70.33±7.99 | 112.29±0.79 | 78.35 |
| | SynthER | 49.76±0.27 | 46.91±0.28 | 91.90±3.75 | 69.21±5.85 | **102.97**±1.65 | 94.08±23.94 | 80.15±16.47 | 90.63±4.66 | 112.12±0.53 | 81.97 |
| | GTA | 54.82±0.35 | 46.89±3.00 | 95.30±0.55 | 77.46±3.42 | 102.11±1.51 | **107.78**±4.66 | 84.40±2.32 | **93.37**±6.35 | 112.87±0.66 | 86.11 |
| | TGCVG | **68.61**±0.39 | **59.63**±0.24 | **95.39**±0.39 | 81.55±5.42 | 100.99±0.37 | 97.30±10.96 | 81.00±2.93 | 86.32±10.94 | **112.99**±0.47 | **87.09** |
| DT | None | 42.43±0.14 | **39.34**±1.22 | 92.43±0.50 | 63.09±2.49 | **81.81**±3.39 | 109.05±2.02 | 71.64±0.69 | 62.06±1.88 | 108.38±0.31 | 74.47 |
| | S4RL | 42.44±0.39 | 38.71±0.83 | 91.80±0.77 | 64.49±1.70 | 66.47±19.27 | **110.57**±0.81 | 72.22±2.49 | 59.67±4.91 | **108.40**±0.24 | 72.75 |
| | GTA | 43.83±0.13 | 37.98±4.97 | 91.78±1.88 | **64.57**±1.22 | 78.43±9.93 | 110.54±0.18 | 74.94±1.72 | 67.90±13.41 | 108.24±0.62 | 75.36 |
| | TGCVG | **62.68**±0.52 | 35.79±9.95 | **92.76**±0.74 | 60.11±4.99 | 73.36±31.59 | 104.87±7.33 | **82.18**±0.99 | **82.96**±3.32 | 108.06±0.59 | **78.09** |

Table 2: Normalized results of TGCVG and baselines on the Maze2D and AntMaze datasets. Average and standard deviation scores are reported over 5 seeds and the best average values are marked in bold. The dataset names are abbreviated as follows: *umaze* to 'u', *medium* to 'm', *large* to 'l', *play* to 'p', and *diverse* to 'd'.

| Algo. | Aug. | maze2d | | | antmaze | | | | | | Average |
|---|---|---|---|---|---|---|---|---|---|---|---|
| | | u | m | l | u | m-p | l-p | u-d | m-d | l-d | |
| IQL | None | 37.41±2.83 | 32.80±1.49 | 58.99±9.16 | 58.75±8.90 | 78.13±3.44 | 40.63±8.75 | 50.38±17.39 | 65.50±9.46 | 45.75±6.34 | 52.04 |
| | S4RL | 37.69±3.36 | 34.82±3.16 | 62.93±3.47 | 55.00±10.47 | 80.88±5.17 | 42.88±8.71 | 51.63±11.67 | 74.00±9.72 | 46.13±8.34 | 53.99 |
| | SynthER | 39.00±2.26 | 34.27±2.51 | 61.74±4.51 | 17.13±6.45 | 41.00±20.58 | 37.50±6.48 | 23.94±11.83 | 40.88±14.15 | 37.50±8.37 | 36.99 |
| | GTA | 41.68±1.41 | 37.78±1.66 | 76.56±4.70 | **66.50**±6.91 | 81.88±4.19 | 44.38±4.66 | **57.88**±9.51 | 78.13±7.85 | 47.75±6.69 | 59.17 |
| | TGCVG | **53.76**±7.50 | **48.41**±3.04 | **103.97**±12.85 | 41.20±4.71 | **83.20**±2.14 | **55.20**±6.68 | 57.20±4.40 | **84.60**±4.32 | **57.20**±4.26 | **64.97** |

## 5.3 ABLATION STUDIES

**How does Transformer-CQL perform compared to the original CQL?** We compare Transformer-CQL with the original CQL, which employs a simple MLP as the policy network. As shown in Table 3, the Transformer-based action synthesizer outperforms the original architecture in CQL, consistent with observations from RCSL methods (Chen et al., 2021; Wang et al., 2024; Hu et al., 2024). We attribute this performance gap to the superior representation capacity of the Transformer backbone.

**How does Transformer-CQL perform compared to Transformer-TD3BC?** As discussed in the method section, we design a conservative value-guided policy for data generation, in contrast to policies constrained by behavior cloning. We compare two augmentation strategies: Transformer-CQL, which represents the conservative Q-value paradigm, and Transformer-TD3BC, a representative method of policy constraint. As shown in Figure 2(a), Transformer-CQL yields stable performance when its augmented data is used to train both CQL and TD3BC. In contrast, Transformer-TD3BC also converges well when paired with TD3BC, but results in a performance collapse when its augmented data is used to train CQL. To further investigate this discrepancy, we visualize the data distributions in Figure 2(b). The figure shows both a portion of the original dataset and the corresponding generated samples. We observe that samples generated by Transformer-CQL remain well-aligned with the original distribution, whereas Transformer-TD3BC produces dense clusters of outlier points (highlighted with green circles). According to (Lyu et al., 2022), policy regularization methods (such as TD3BC) typically exert less constraint on Q-values than value-penalization methods (such as CQL). As a result, Transformer-TD3BC may generate actions that fall outside the support of the original dataset. In the subsequent offline RL training phase, these dense outlier

Table 3: Ablation on the action synthesizer. The final scores are trained with TD3BC. Average and standard deviation scores are reported over 5 seeds. The best average values are marked in bold.

| Action Synthesizer | hopper-medium | walker2d-medium | halfcheetah-medium |
|---|---|---|---|
| CQL | 60.66±3.83 | 84.98±0.82 | 48.43±0.32 |
| Transformer-CQL | **90.47**±4.06 | **87.50**±0.20 | **68.14**±0.58 |

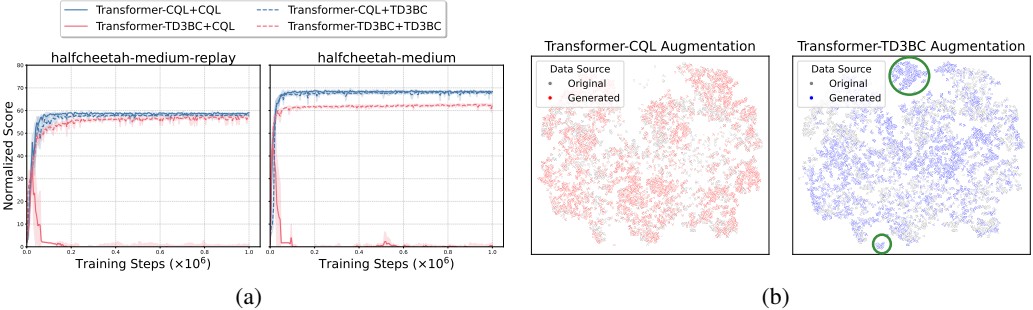

Figure 2: (a) Ablation study comparing Transformer-CQL and Transformer-TD3BC. (b) t-SNE visualization of generated data distributions on *halfcheetah-medium*. Each point represents a state-action pair.

clusters are incorrectly treated as in-distribution data due to CQL's conservatism, while the original sparse samples are mistakenly penalized as OOD data. This mismatch leads to inaccurate value estimation and poor performance.

**The effect of $\lambda$.** The tradeoff coefficient $\lambda$ plays a critical role in value penalization. We conduct experiments by varying $\lambda$ on different datasets to investigate how the level of conservatism affects performance. As shown in Figure 3(a), changing $\lambda$ has limited impact on the *walker-medium* dataset but significantly influences performance on *hopper-medium*. A smaller $\lambda$, corresponding to a lower degree of conservatism, tends to improve the performance of the downstream offline algorithms. Figure 3(b) further illustrates that the effect of $\lambda$ manifests through its influence on the decision-making capacity of Transformer-CQL, which in turn determines the quality of the synthesized data and indirectly impacts the final offline learning performance. This insight provides a practical guideline for applying our TGCVG: the quality of synthesized trajectories is highly dependent on the decision-making ability of the action synthesizer. Therefore, improving the policy model's capability during the early stage of training can lead to higher-quality data generation.

## 5.4 FURTHER ANALYSIS

**Data Quality Analysis.** We adopt the data quality metrics introduced in GTA (Lee et al., 2024) to evaluate our generated datasets. Specifically, we consider three metrics: (1) novelty, which quantifies the ability of an augmentation method to explore novel state-action pairs; (2) optimality, which measures the actual rewards of the generated data; and (3) dynamic MSE, which computes the normalized discrepancy between synthesized and real states after applying the transition dynamics. Although baselines provide code for data generation, the implementations of the evaluation metrics are not publicly available. To ensure a fair and consistent comparison, we reproduce the datasets using the official codebase of each method and evaluate all datasets using our own implementation of the evaluation pipeline. Table 4 reports the results on standard Gym tasks. While our TGCVG does not achieve the best score in every individual metric, it consistently yields the best overall performance. Compared to GTA, our TGCVG achieves better dynamic MSE despite slightly lower optimality, which may imply that dynamic consistency is a more important prerequisite than reward magnitude when evaluating the quality of generated data. Notably, although our TGCVG constrains trajectory generation within the original data distribution via value penalization, the novelty scores indicate that the generated samples are not mere replicas of existing ones. This suggests that the

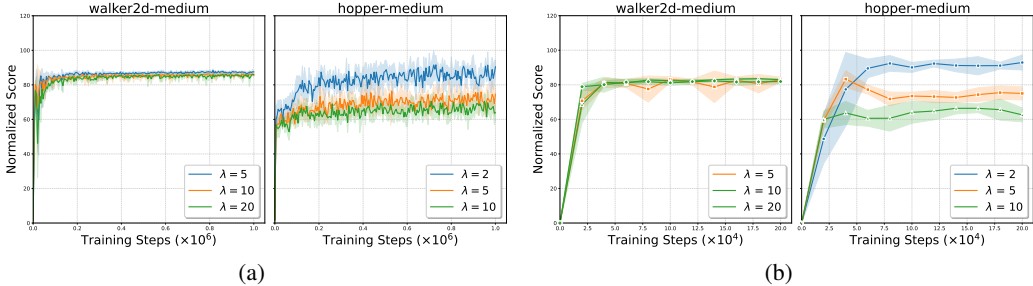

(a)                                          (b)

Figure 3: Ablation on the $\lambda$. (a) Final performance of TD3BC trained on our augmented datasets. Average and standard deviation scores are reported over 5 seeds. (b) Evaluation scores of the Transformer-CQL policy networks, before interacting with dynamics models to generate synthetic trajectories. Average and standard deviation scores are reported over 5 seeds.

Table 4: Data quality analysis of TGCVG and baselines across Gym locomotion tasks. Arrows ↑/↓ indicate whether higher/lower values are better. The best and the second-best results of each setting are marked as **bold** and underline, respectively.

| Task | Novelty (↑) | | | | Optimality (↑) | | | | Dynamics MSE (↓) | | | |
|---|---|---|---|---|---|---|---|---|---|---|---|---|
| | S4RL | Synther | GTA | TGCVG | S4RL | Synther | GTA | TGCVG | S4RL | Synther | GTA | TGCVG |
| halfcheetah-medium | 0.00 | 3.72 | 4.21 | **5.80** | 4.77 | 4.78 | **6.26** | 6.25 | **0.05** | 2.04 | 2.64 | 1.29 |
| halfcheetah-medium-replay | 0.00 | 10.91 | 9.47 | **13.68** | 3.09 | 3.10 | 3.21 | **4.70** | **0.05** | 4.82 | 8.05 | 4.80 |
| halfcheetah-medium-expert | 0.00 | 3.16 | **6.00** | 3.63 | 7.71 | 7.69 | **9.39** | 8.06 | **0.03** | 2.54 | 5.57 | 0.86 |
| hopper-medium | 0.00 | 0.09 | 0.10 | **0.11** | 3.11 | 3.11 | **3.74** | 3.22 | **0.11** | 0.22 | 0.18 | 0.13 |
| hopper-medium-replay | 0.00 | 0.32 | 0.19 | **0.46** | 2.37 | 2.37 | **2.59** | 2.57 | **0.09** | 0.24 | 0.10 | 0.12 |
| hopper-medium-expert | 0.00 | 0.08 | **0.11** | 0.08 | 3.36 | 3.35 | **3.84** | 3.48 | **0.08** | 0.13 | 0.10 | **0.08** |
| walker2d-medium | 0.00 | 0.84 | 1.07 | **1.12** | 3.39 | 3.40 | **3.61** | 3.46 | 4.27 | 4.34 | 5.87 | **4.24** |
| walker2d-medium-replay | 0.00 | 3.49 | 2.53 | **3.64** | 2.47 | 2.46 | **2.74** | 2.65 | 7.92 | 8.73 | 8.90 | **6.33** |
| walker2d-medium-expert | 0.00 | 0.62 | **0.84** | 0.77 | 4.16 | 4.17 | **4.29** | 4.27 | 4.62 | 4.68 | 5.07 | **4.35** |

interaction between the policy and dynamics models enables a degree of generalization beyond the training data.

**Training and Generation Time Comparison for TGCVG and GTA.** We compare the training time of our TGCVG and GTA, the Diffusion-based method guided by scaled returns. All experiments are conducted on a single NVIDIA RTX TITAN GPU using the *halfcheetah-medium-v2* dataset, with $2 \times 10^5$ training steps and $5 \times 10^6$ synthesized data points. As shown in Figure 4, our TGCVG enables parallel training of the dynamics model and the policy generation model, leading to significantly reduced training time compared to the diffusion process in GTA. Moreover, our TGCVG achieves much faster data generation by avoiding the time-consuming partial noising and denoising stages required by GTA. These results demonstrate the substantial reduction in time cost achieved by our approach.

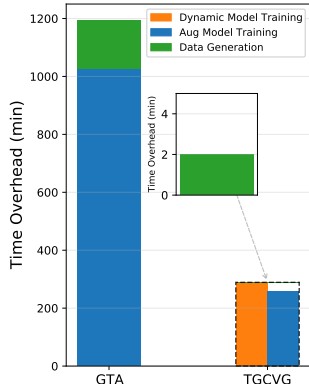

Figure 4: Compare the time overhead of our TGCVG and GTA.

## 6 CONCLUSION

We propose Trajectory Generation with Conservative Value Guidance (TGCVG), a simple yet effective data augmentation framework for offline RL. The Q-learning module encourages above-average actions compared to the dataset, while value-based penalization constrains generated samples within the data distribution, mitigating OOD errors. Experiments on standard offline benchmarks show that our TGCVG not only outperforms state-of-the-art baselines but also significantly reduces computational overhead. These results demonstrate the potential of leveraging the interaction between an offline policy and a learned dynamics model to synthesize high-quality data, mimicking the data collection process in online RL.

## 7 Reproducibility Statement

We have made every effort to ensure the reproducibility of our work. A detailed description of the proposed method is provided in Section 4, and all hyperparameter settings are included in Appendix A.1. The datasets used in our experiments are publicly available through the D4RL benchmark (Fu et al., 2020).

### Acknowledgments

This work is supported in part by the Guangxi Key Research and Development Program under Grant FN2504240005; in part by the National Natural Science Foundation of China under Grant 62471064; in part by the Fundamental Research Funds for the Beijing University of Posts and Telecommunications under Grant 2025AI4S02.

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

# A  EXPERIMENTAL DETAILS

## A.1  DETAILS OF TGCVG

We implement TGCVG based on the official GTA codebase (`https://github.com/Jaewoopudding/GTA`), replacing the original Diffusion module with our Transformer-based synthesizer. For policy network training, most hyperparameters of the Transformer backbone follow those used in DT (Chen et al., 2021), as detailed in Table 5. For the context length $L$ on the Gym domain, we set $L = 5$ for *halfcheetah-medium* and *halfcheetah-medium-replay*, and $L = 20$ for the remaining tasks, following (Hu et al., 2024). On the Maze2D and AntMaze domains, we consistently set $L = 5$. Our conservative Q-value guidance is adapted from CQL (Kumar et al., 2020), with modifications to the tradeoff coefficient $\lambda$ to balance conservatism and exploration. During the data augmentation phase, we adopt a similar pipeline to GTA, and the dynamics model is implemented using the official ADMPO codebase (`https://github.com/LAMDA-RL/ADMPO`). The sample length $K$ is chosen based on the complexity of each environment. The generated dataset includes 5 million transitions for each task. Complete hyperparameter settings for all evaluated tasks are summarized in Table 6.

Table 5: Hyperparameters of TGCVG in our experiments.

| Parameter | Value |
|---|---|
| Number of layers | 3 |
| Number of attention heads | 1 |
| Embedding dimension | 256 |
| Nonlinearity function | GeLU |
| Batch size | 256 |
| Dropout | 0.1 |
| Optimizer | Adam (Kingma, 2014) |
| Learning rate | 1e-4 |

Table 6: Hyperparameter settings of all datasets.

| Datasets | $\lambda$ | $K$ |
|---|---|---|
| halfcheetah-medium-v2 | 0.1 | 60 |
| halfcheetah-medium-replay-v2 | 0.1 | 60 |
| halfcheetah-medium-expert-v2 | 10.0 | 60 |
| hopper-medium-v2 | 2.0 | 60 |
| hopper-medium-replay-v2 | 2.0 | 40 |
| hopper-medium-expert-v2 | 5.0 | 60 |
| walker2d-medium-v2 | 5.0 | 40 |
| walker2d-medium-replay-v2 | 5.0 | 40 |
| walker2d-medium-expert-v2 | 10.0 | 40 |
| maze2d-umaze-v1 | 0.5 | 7 |
| maze2d-medium-v1 | 0.3 | 7 |
| maze2d-large-v1 | 0.4 | 7 |
| antmaze-umaze-v2 | 5 | 6 |
| antmaze-medium-play-v2 | 3 | 7 |
| antmaze-large-play-v2 | 1 | 7 |
| antmaze-umaze-diverse-v2 | 5 | 7 |
| antmaze-medium-diverse-v2 | 1 | 7 |
| antmaze-large-diverse-v2 | 2 | 7 |

## A.2 Offline RL Algorithms

We utilize the Clean Offline Reinforcement Learning (CORL) library (Tarasov et al., 2023) as the implementation base for all offline RL algorithms, available at (`https://github.com/tinkoff-ai/CORL`). For the Gym domain, we adopt the default hyperparameters for CQL, IQL, TD3+BC, and DT. For the Maze2D and AntMaze tasks, IQL hyperparameters are configured according to the original paper.

## A.3 Dataset and Evaluation

We use the v2 datasets for the Gym locomotion and AntMaze tasks, and the v1 datasets for Maze2D, consistent with GTA. For all tasks evaluated, we train TD3+BC, IQL, and CQL for 1 million steps and report performance based on final evaluations. For DT, we train for 200,000 steps due to its faster convergence. We report final evaluation scores averaged over 10 episodes for Gym locomotion tasks and 100 episodes for Maze2D and AntMaze tasks. All scores are normalized following the D4RL protocol (Fu et al., 2020), where a score of 0 corresponds to a random policy and 100 corresponds to an expert policy.

# B Additional Experiments

## B.1 Ablation Study on Return Condition

Transformer-based methods in offline RL typically formulate the task as a sequence modeling problem, using the RTG as a condition to generate high-return trajectories. Recent approaches such as QT (Hu et al., 2024) incorporate Q-learning into the Transformer architecture and have shown effective policy improvement. However, we investigate whether the RTG input remains necessary when the Transformer is already equipped with a Q-learning module. To this end, we conduct an ablation study on RTG using the official QT codebase (`https://github.com/charleshsc/QT`). As shown in Table 7, removing RTG has little impact on performance, suggesting that the Q-learning component is the primary driver of policy improvement. Based on this observation, we design our Transformer-CQL model to take only state sequences as input, which reduces input tokens and improves computational efficiency.

Table 7: Ablation on RTG conditioning in QT. Average and standard deviation scores are reported over 3 seeds. The best average values are marked in bold.

| Dataset | QT w/o RTG | QT w/ RTG |
|---|---|---|
| hopper-medium | 73.29$\pm$1.15 | **77.28**$\pm$4.07 |
| hopper-medium-replay | 99.24$\pm$0.23 | **100.99**$\pm$0.51 |

## B.2 Ablation Study on Sample Length

We investigate the effect of sample length $K$ in the data synthesis process. As shown in Table 8, increasing $K$ improves trajectory coverage and allows the policy model to explore longer, higher-return behaviors, particularly in stable environments such as *halfcheetah*. However, a larger $K$ also increases the likelihood of sampling trajectories that contain terminal states. Since our method inherits terminal flags from the original trajectories, this can result in inconsistencies: the generated portion may not actually reach a terminal condition, but the trajectory is still forcibly terminated. Such mismatches between true and inherited termination become more pronounced as $K$ increases, especially in environments with well-defined terminal conditions like *hopper*. Conversely, smaller values of $K$ reduce the chance of terminal mismatch but may lead to insufficient trajectory exploration, as the generated segment is shorter and less likely to achieve high returns. These results highlight a tradeoff in choosing the appropriate sample length $K$ during synthesis.

Table 8: Ablation on the number of sampled steps $K$. The final scores are trained with TD3BC. Average and standard deviation scores are reported over 5 seeds. The best average values are marked in bold.

| Dataset | $K = 25$ | $K = 40$ | $K = 60$ | $K = 80$ | $K = 100$ |
|---|---|---|---|---|---|
| halfcheetah-medium | 65.69±0.60 | 66.99±0.72 | 68.14±0.58 | 67.42±2.11 | **69.21**±0.30 |
| hopper-medium | 91.06±4.07 | **93.94**±4.60 | 90.47±4.06 | 86.50±5.02 | 87.24±4.66 |

### B.3 ABLATION STUDY ON ORIGINAL DATA SIZE

To evaluate the robustness of our method in low-data regimes, we conduct experiments using only 1% and 10% of the original trajectories from the D4RL datasets. As shown in Table 9, our method demonstrates exceptional data efficiency. Notably, on *halfcheetah-medium*, TGCVG achieves a score of 49.90 using only 1% of the data, surprisingly surpassing the original TD3BC trained on the full dataset (100%) (48.42). Furthermore, our method exhibits superior scaling capabilities compared to the strongest augmentation baseline, GTA. While both methods perform comparably on *hopper-medium* at 1% data, TGCVG achieves a massive performance jump at 10% data (reaching 86.58), whereas GTA stagnates (58.16). This highlights the effectiveness of our generative augmentation in extracting and amplifying useful patterns even from extremely limited data.

Table 9: Ablation study on the size of the original dataset. We compare our method against the strongest augmentation baseline (GTA) and the original TD3BC algorithm (No Aug). The final scores are trained with TD3BC. Average and standard deviation scores are reported over 5 seeds. The best average values are marked in bold.

| Dataset | Method | 1% Data | 10% Data | 100% Data (Full) |
|---|---|---|---|---|
| halfcheetah-medium | Original (No Aug) | $9.26 \pm 7.90$ | $48.38 \pm 0.21$ | $48.42 \pm 0.62$ |
| | GTA | $46.24 \pm 0.11$ | $53.21 \pm 0.23$ | $57.84 \pm 0.51$ |
| | **TGCVG (Ours)** | **49.90** $\pm$ 0.23 | **57.00** $\pm$ 0.30 | **68.14** $\pm$ 0.58 |
| hopper-medium | Original (No Aug) | $35.57 \pm 2.28$ | $54.98 \pm 3.94$ | $61.04 \pm 3.18$ |
| | GTA | **56.56** $\pm$ 3.64 | $58.16 \pm 7.97$ | $69.57 \pm 4.05$ |
| | **TGCVG (Ours)** | $54.04 \pm 12.13$ | **86.58** $\pm$ 5.13 | **90.47** $\pm$ 4.06 |

### B.4 ABLATION STUDY ON SYNTHETIC DATA SIZE

We further investigate the impact of the volume of generated synthetic data on downstream performance. We vary the synthetic dataset size from 0.01M to 5M transitions. Table 10 presents the comparison between our TGCVG and the GTA baseline. The results indicate that our method is highly data-efficient. On *halfcheetah-medium*, TGCVG trained with only 0.1M synthetic data (63.08) already outperforms GTA trained with 5M data (57.84). Furthermore, our method exhibits superior scalability: while GTA's performance tends to plateau or increase slowly, TGCVG shows a continuous and substantial performance gain as the synthetic data size increases, particularly on *hopper-medium*, where it reaches a score of 90.47 with 5M samples.

### B.5 PERFORMANCE COMPARISON BETWEEN TGCVG AND MODEL-BASED METHODS

Model-based methods also generate data, though their data generation is typically not decoupled from the learning process, unlike data augmentation approaches. We are interested in whether data generated by model-based methods can be repurposed to train offline model-free algorithms for performance improvement. Since our dynamics model is adapted from ADMPO (Lin et al., 2024), we compare our method with ADMPO to investigate this question. Table 11 compares ADMPO+IQL, which reuses data from ADMPO training for IQL, with the original ADMPO scores and our TGCVG results. The results suggest that data generated by model-based methods is not well-suited for direct use in offline model-free training, demonstrating that TGCVG provides a more general and effective data generation strategy for offline RL.

Table 10: Ablation study on the size of the synthetic dataset (generated transitions). We compare the scalability of our method versus GTA. The final scores are trained with TD3BC. Average and standard deviation scores are reported over 5 seeds. The best average values are marked in bold.

| Dataset | Method | 0.01M | 0.1M | 1M | 5M (Default) |
|---|---|---|---|---|---|
| halfcheetah-medium | GTA | $49.46 \pm 0.28$ | $54.03 \pm 0.45$ | $57.62 \pm 0.37$ | $57.84 \pm 0.51$ |
| | TGCVG | $\mathbf{51.21} \pm 1.03$ | $\mathbf{63.08} \pm 1.85$ | $\mathbf{67.92} \pm 0.64$ | $\mathbf{68.14} \pm 0.58$ |
| hopper-medium | GTA | $57.00 \pm 1.64$ | $56.36 \pm 3.13$ | $63.91 \pm 2.88$ | $69.57 \pm 4.05$ |
| | TGCVG | $\mathbf{59.00} \pm 3.02$ | $\mathbf{60.53} \pm 4.47$ | $\mathbf{66.08} \pm 5.38$ | $\mathbf{90.47} \pm 4.06$ |

Table 11: Normalized results on *halfcheetah-medium* and *antmaze-large-diverse*. Average and standard deviation scores are reported over 5 seeds. The best average values are marked in bold.

| Algo. | halfcheetah-medium | antmaze-large-diverse |
|---|---|---|
| ADMPO+IQL | $21.90 \pm 3.91$ | $0.00 \pm 0.00$ |
| ADMPO | $\mathbf{72.20} \pm 0.60$ | $0.00 \pm 0.00$ |
| TGCVG+IQL | $68.61 \pm 0.39$ | $\mathbf{57.20} \pm 4.26$ |

## B.6 FURTHER STUDY ON COMPOUNDING ERROR

Compounding errors remain a major challenge for model-based methods in offline RL. To assess this issue, we follow the evaluation protocol used in ADMPO (Lin et al., 2024) and plot the prediction error curves over rollout steps. We compare TGCVG and GTA using generated trajectories with rollout lengths corresponding to those used in ADMPO. For a fair comparison, we evaluate prediction errors over the shared rollout horizon, defined as the minimum trajectory length between the two methods. As shown in Figure 5, the prediction errors of TGCVG and GTA remain close and increase slowly with rollout length, indicating that the dynamic plausibility of the generated sequences is well preserved.

## B.7 FURTHER STUDY ON DATA QUALITY

We further analyze how data quality evolves with increasing rollout length. As shown in Figure 6, both novelty and optimality metrics improve as the rollout horizon increases. This suggests that the policy network becomes less dependent on the original dataset and gradually shifts toward generating more diverse and higher-reward samples. Meanwhile, the dynamic MSE remains relatively stable across different rollout lengths, indicating that the interaction between the policy and dynamics models consistently produces transitions that are dynamically plausible.

## C HYPERPARAMETER TUNING GUIDELINES

We provide a principled guideline for selecting the conservatism weight $\lambda$ and the generation length $K$. To address concerns regarding the search space, we propose a systematic "Initialization + Local Fine-tuning" workflow that leverages established community knowledge to minimize tuning overhead.

**Conservatism Weight ($\lambda$): A Systematic Workflow.** Instead of a global search, our selection of $\lambda$ inherits heuristics from the backbone algorithm (CQL) and aligns with the "BC + Q-value Regularization" paradigm (e.g., TD3BC (Fujimoto & Gu, 2021)). We recommend the following two-step process:

1. **Initialization (Anchor):** Since our method builds upon the CQL backbone, practitioners can initialize $\lambda$ using standard settings recommended in the original CQL literature (e.g., $\lambda \approx 5.0$ or $10.0$ as a starting baseline).

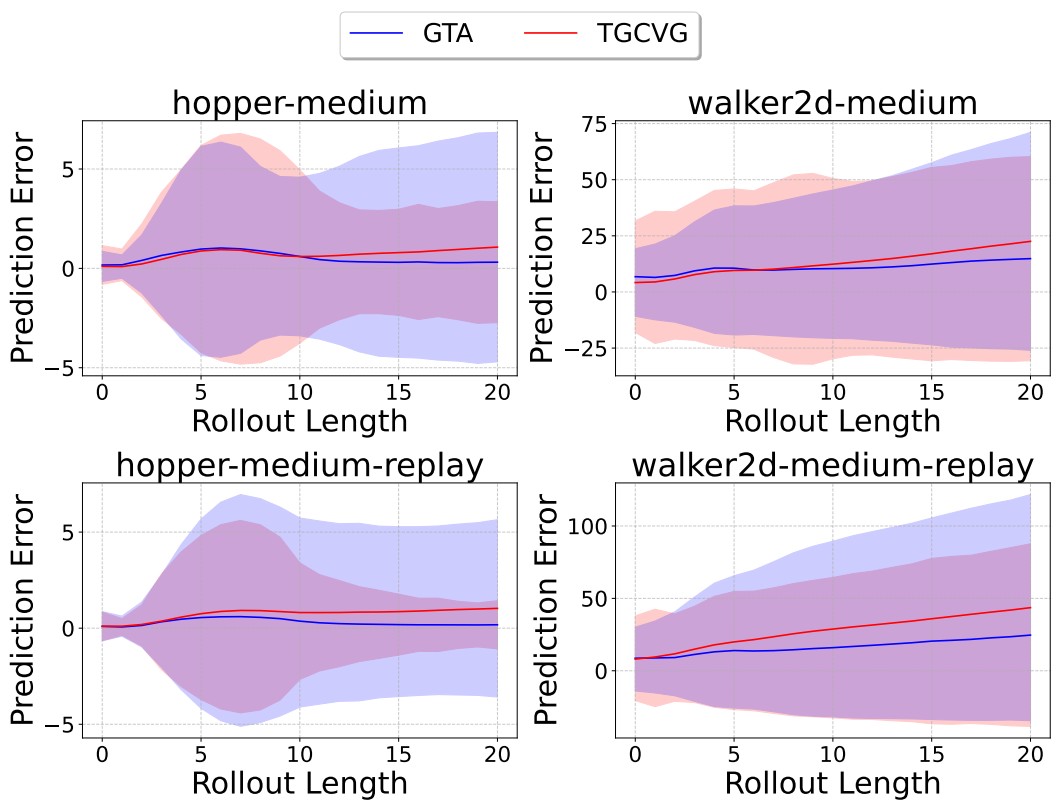

Figure 5: Comparison among TGCVG and GTA, in terms of the growth curve of the compounding error as rollout length increases, after data generation.

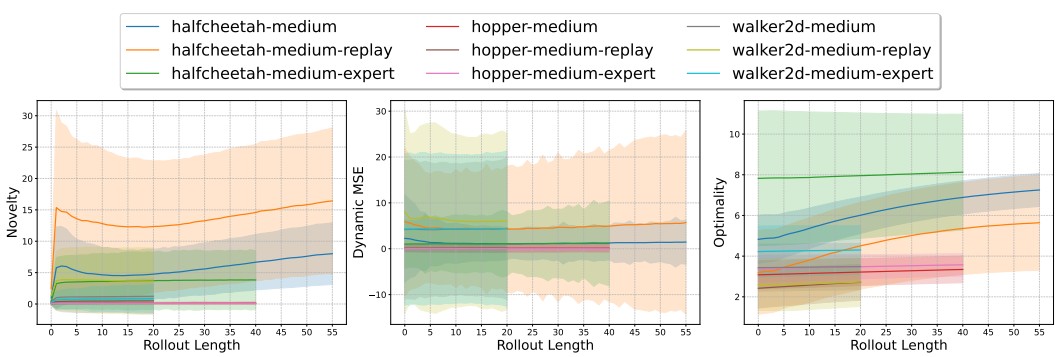

Figure 6: Data quality analysis of TGCVG as rollout length increases.

2. **Adjustment (Return-Distribution-Aware Tuning):** We then refine this anchor based on the return distribution of the dataset, following an inverse logic to TD3BC:

   - **Unimodal / Concentrated Distributions:** For datasets concentrated around a single return mode (e.g., *medium* or *medium-replay*), we adjust $\lambda$ to be lower. A reduced conservative penalty allows the policy to leverage Q-value guidance for extrapolation, improving upon the average behavior.

   - **Multimodal Distributions:** For datasets with conflicting modes (e.g., *medium-expert*), we adjust $\lambda$ to be higher. Stronger conservatism is necessary to restrict the

policy to high-density regions, preventing invalid transitions that interpolate between distinct behavioral modes.

3. **Local Fine-tuning:** Finally, practitioners only need to perform a minimal search in the local neighborhood of the adjusted anchor, significantly reducing the search space compared to a blind sweep.

**Generation Length ($K$): Termination-Aware Selection.** The choice of $K$ is deterministic based on the strictness of the environment's termination conditions:

- **Lenient Termination:** In environments where early termination is rare (e.g., *halfcheetah*), a longer $K$ is preferred. This enables the policy to explore more extensively and stitch together longer, higher-quality trajectories.
- **Strict Termination:** In environments with strict termination conditions or sparse rewards (e.g., *antmaze*), a shorter $K$ is crucial. This prevents "terminal mismatch", a scenario where a high-quality generated trajectory might trigger a premature termination in the simulator that is inconsistent with the original long-horizon plan.

## D    LIMITATIONS

Our work proposes a novel data augmentation framework for offline RL that improves the quality of generated trajectories while reducing computational overhead. However, since our method builds upon the implementation of CQL, it inherits certain limitations of CQL itself. In particular, tasks that are inherently challenging for CQL remain difficult for our Transformer-CQL variant, even though our policy model is more robust and expressive. When the policy fails to produce high-quality actions, the benefit of incorporating generated data into offline RL training is limited.

## E    FUTURE WORK

Several promising directions remain for future exploration. First, iterative grounding mechanisms inspired by tree-based key frame extraction (Cao et al., 2025) could enable selective augmentation of high-value trajectory segments. Second, unsupervised anomaly detection (Qiu et al., 2025) may filter low-quality synthetic trajectories. Finally, service-oriented architectures (Duan et al., 2025) would facilitate flexible deployment across diverse robotic platforms.

## F    THE USE OF LARGE LANGUAGE MODELS (LLMS)

In accordance with the ICLR policy on the use of large language models (LLMs), we disclose that LLMs are employed during the preparation of this paper. Their use is limited to assisting with language editing, improving readability, and polishing the presentation of the manuscript. LLMs are not involved in formulating the research questions, designing the methodology, conducting experiments, analyzing results, or drawing scientific conclusions. We take full responsibility for all technical content, claims, and conclusions presented in this work.

