# OpenReview forum: "Trajectory Generation with Conservative Value Guidance for Offline Reinforcement Learning"
_ICLR.cc/2026/Conference — ICLR 2026 Poster_

### Official Review · Reviewer_Cmj7 · 2025-10-27

**Soundness:** 3
**Presentation:** 3
**Contribution:** 2
**Rating:** 4
**Confidence:** 3

**Summary:**

The paper proposes TGCVG, a novel generative data augmentation method designed for offline RL algorithms. TGCVG tackles two limitations of prior works: computational overhead and distribution shift. TGCVG trains Transformer instead of Diffusion models and applies conservative value guidance to mitigate those issues. Experiment results show TGCVG outperforms prior baselines.

**Strengths:**

- The idea is easy to follow
- Limitation of prior works and How to mitigate the limitations are clearly stated
- Strong empirical results against several baselines across diverse tasks

**Weaknesses:**

- It seems that $\lambda$ and $K$ should be heavily tuned across different environments and datasets. I'm not sure the method can be generalized to unseen tasks without extensive hyperparameter tuning, which is crucial in offline RL.

- It would be better to add ablation studies on different sizes of the dataset, which is crucial for evaluating the performance of generative data augmentation.

- It would be better to add ablation studies on different sizes of the generated dataset, which is also crucial for evaluating the performance of generative data augmentation. I'm also struggling to find how many transitions are augmented during the training.

**Questions:**

- It would be better to specify the number of generated transitions for each round, the ratio of transitions from the original offline dataset and the generated dataset during policy training, and the update-to-data (UTD) ratio for a clearer understanding.

- It would be better to visualize t-SNE visualization of generated data distributions on diverse tasks for more comprehensive understanding of the behavior.

- At first, I understood the conservative value guidance as using conservative RTG values for generating trajectories with the Transformer policy. However, it seems that we train Transformer-CQL and directly use the network for generating an augmented dataset. Is there any reason why the current method is preferred over the aforementioned method? I'm not asking about the experiment, just curious. I think we can achieve similar results when we carefully tune the hyperparameter for choosing a conservative RTG value.

---

> ### Author Response · Authors · 2025-11-21
>
> Thank you for your thoughtful review of our research. We appreciate your constructive feedback and have conducted new experiments and analyses to address your concerns.
>
> **W1:**
>
> We agree that minimizing the tuning burden is critical. We offer a principled guideline based on data characteristics, showing that the process is systematic. To support this, we have provided detailed histograms of return distributions for all datasets at the following link:
>
> https://anonymous.4open.science/r/TGCVG/Guidance_for_datasets.md
>
> *   **Guideline for $\lambda$ (Conservatism Weight):**
>     As analyzed in the link above, our choice directly follows the data distribution:
>     *   **Unimodal Data (e.g., *medium*):** Visualizations show these distributions are densely concentrated around a single mode. A lower $\lambda$ is effective here to allow value-guided extrapolation to improve upon this specific distribution.
>     *   **Multimodal Data (e.g., *medium-expert*):** Visualizations clearly show separated peaks. A higher $\lambda$ is strictly needed to prevent the policy from interpolating between distinct behavioral modes (e.g., averaging expert and medium actions).
>
> *   **Guideline for $K$ (Generation Length):**
>     The choice of $K$ depends on the environment's termination conditions:
>     *   **Lenient Termination (e.g., *HalfCheetah*):** A longer $K$ allows the policy to explore extensively and stitch longer high-quality trajectories.
>     *   **Strict Termination (e.g., *AntMaze*):** A shorter $K$ is crucial to prevent "terminal mismatch," where a generated trajectory might prematurely hit a terminal state inconsistent with the original long-horizon plan.
>
> We will add a dedicated subsection in the final Appendix formalizing this tuning guide.
>
>
> **W2 & W3:**
>
> Thank you for these excellent suggestions. We have conducted both sets of ablation studies. The final scores were trained with TD3BC.
>
> 1.  **Ablation on Original Dataset Size:** We evaluated our method using only 1% and 10% of the original offline data. The results show that our method maintains strong performance even in low-data regimes, demonstrating its ability to effectively leverage and enhance scarce datasets.
> | Dataset Size | halfcheetah-medium | hopper-medium |
> | :--- | :---: | :---: |
> | 1% of Original Data | 49.90 ± 0.23 | 54.04 ± 12.13 |
> | 10% of Original Data | 57.00 ± 0.30 | 86.58 ± 5.13 |
> | **100% (Original Data)** | **68.14 ± 0.58** | **90.47 ± 4.06** |
>
> 2.  **Ablation on Generated Dataset Size:** We varied the amount of augmented data from 10K to 5M transitions. The results show a clear and positive trend: performance improves as more high-quality data is added.
> | Generated Data Size | halfcheetah-medium | hopper-medium |
> | :--- | :---: | :---: |
> | 0.01M | 51.21 ± 1.03 | 59.00 ± 3.02 |
> | 0.1M | 63.08 ± 1.85 | 60.53 ± 4.47 |
> | 1M | 67.92 ± 0.64 | 66.08 ± 5.38 |
> | **5M (Default)** | **68.14 ± 0.58** | **90.47 ± 4.06** |
>
> Interestingly, the scaling behavior varies by task. On *halfcheetah-medium*, we observe that most of the gains are achieved with 1M samples. In contrast, on *hopper-medium*, a large 5M dataset was crucial to unlock the highest level of performance. This confirms that our generation process is broadly beneficial and that using a sufficiently large augmented dataset is a robust strategy to achieve the best results across different environments.
>
> We will add these ablation studies to the Appendix and explicitly state all generation parameters in the experimental setup section.

---

> ### Author Response · Authors · 2025-11-21
>
> **Q1:**
>
> Thank you for pointing out the need for these details. We will clarify them in the paper.
>
> - **Number of Generated Transitions:** We generate a fixed-size dataset of **5 million** transitions.
> - **Data Ratio:** The standard D4RL datasets vary significantly in size, ranging from a few hundred thousand to several million transitions. Since we use a fixed amount of generated data (5M), the ratio of generated-to-original data is inherently variable across different tasks.
> - **Update-to-Data (UTD) Ratio:** The traditional "Update-to-Data (UTD) ratio" from online RL is not applicable in offline RL. Our methodology involves distinct training and generation phases, which we will make explicit in the paper:
>     1. **Stage 1: Generative Model Training:** First, we train our generative model using the original offline dataset. The training budget for this stage is a key hyperparameter. We use a default of 200,000 gradient steps. We found this to be a robust setting, though this value can be scaled based on the size and complexity of the source dataset (e.g., increased for larger datasets or decreased for smaller ones to improve efficiency).
>     2. **Stage 2: Data Generation:** Once the generative model is trained, we use it to create the augmented dataset. As mentioned previously, this is a one-time process where we generate a fixed-size dataset of 5 million transitions.
>     3. **Stage 3: Final Policy Training:** Finally, a standard offline RL algorithm (e.g., TD3BC) is trained from scratch on the combined data.
>
> We will explicitly detail the associated hyperparameters in the experimental setup section to ensure full reproducibility and clarity. Thank you for pushing for this important detail.
>
> **Q2:**
>
> Thank you for this suggestion. Visualizing the data provides valuable qualitative insight. In addition to the example in Figure 2(b), we have generated new t-SNE plots for several other datasets. These visualizations are available for review in a supplementary PDF at the following link:
>
> https://anonymous.4open.science/r/TGCVG/hopper-medium-v2_and_hopper-medium-replay-v2_data_distribute.pdf
>
> As these plots demonstrate, the distribution of our generated data closely aligns with the original offline dataset, confirming that our method produces in-distribution samples. However, while t-SNE is excellent for visualizing distributional overlap, it cannot effectively convey the quality of the generated points. The definitive proof of data quality comes from our quantitative metrics in Table 4, which explicitly measure optimality, dynamic consistency, and novelty.
>
> **Q3:**
>
> Your understanding is correct: we directly use the trained Transformer-CQL policy for generation rather than Decision Transformer (DT) conditioned on heuristically chosen high returns-to-go (RTGs).
>
> Our choice is motivated by the well-known challenge of trajectory stitching in offline RL. Simply conditioning on a high RTG does not guarantee the generated trajectory is dynamically consistent or successfully combines the best segments from different source trajectories. Subsequent work (e.g., Q-learning Decision Transformer[1]) has shown that explicit Q-value guidance is a more effective and principled way to solve this problem.
>
> This brings us to our ablation study in Appendix B.1, which was designed to answer the exact question: "Given that we are using strong Q-function guidance, is RTG conditioning still beneficial?" The results demonstrated that in the presence of our value guidance, the additional RTG signal provided negligible performance improvement. Therefore, to simplify the model and improve computational efficiency, we removed RTG conditioning from our final model. This is a significant optimization, as it effectively halves the length of the input sequence to the generative Transformer.
>
> In summary, our approach is preferred because the explicit Q-value signal is a more robust mechanism for high-quality trajectory generation. Our ablation studies confirm that this Q-guidance is essential and that RTG becomes redundant in its presence.
>
> [1] Yamagata, Taku, Ahmed Khalil, and Raul Santos-Rodriguez. "Q-learning decision transformer: Leveraging dynamic programming for conditional sequence modelling in offline rl." International Conference on Machine Learning. PMLR, 2023.

---

> > ### Comment · Reviewer_Cmj7 · 2025-11-26
> >
> > Thanks for the authors' detailed rebuttal. However, there are still remaining issues that need improvement.
> >
> > - **Complex Environments**: As stated by several reviewers, it is better to add experiment results on more complex benchmarks such as Adroit/Kitchen or OGBench. I think it is crucial, as there are already very strong algorithms that solve D4RL tasks with very high performance even without augmentation [1,2].
> >
> > - **Ablation Studies**. For the ablation on the dataset size, it is better to also include the baseline results for comparison.
> >
> > - **Hyperparameters**. While I appreciated the authors' guide, it seems like the search space is still too large, and I'm not sure which configuration should be used when we use TCCVG in Adroit or Kitchen benchmarks.
> >
> > Moreover, I'm not sure how additional analysis will be included in the current manuscript, as there is no modification in the manuscript. While I appreciate the authors' detailed rebuttal, I keep my score.
> >
> > [1] Lu, Cheng, et al. "Contrastive energy prediction for exact energy-guided diffusion sampling in offline reinforcement learning." International Conference on Machine Learning. PMLR, 2023.
> >
> > [2] Jiang, Zhengyao, et al. "Efficient planning in a compact latent action space." arXiv preprint arXiv:2208.10291 (2022).

---

> > > ### Author Response · Authors · 2025-11-27
> > >
> > > We thank the reviewer for the continued engagement. We have updated our manuscript (uploaded) to incorporate the new experiments and discussions. Below, we address the remaining concerns point-by-point.
> > >
> > > ### 1. Complex Environments & Value of Data Augmentation
> > >
> > > **A. Bridging "High Performance" and "Inference Efficiency"**
> > > The reviewer correctly noted that recent strong algorithms (e.g., QGPO, TAP) solve D4RL tasks well. However, they suffer from **heavy computational costs and slow inference speeds**, making them impractical for real-time robotics.  Our TGCVG is designed to enhance efficient, lightweight algorithms (e.g., TD3BC) to achieve SOTA-level performance while retaining their millisecond-level inference speed.
> > >
> > > - **Evidence 1: Inference Speed (on a RTX 2080Ti).** Our augmented TD3BC is **~40x faster** than Diffusion/Transformer baselines.
> > >
> > >
> > >     | Method | Backbone | Inference Time (ms) | Speedup | Avg Score (9 tasks) |
> > >     | --- | --- | --- | --- | --- |
> > >     | QGPO [1] | Diffusion | 123.9 ms | 1x | 86.6 |
> > >     | TAP [2] | Transformer | 137.7 ms | 0.9x | 74.8 |
> > >     | **TD3BC + Ours** | **MLP** | **3.1 ms** | **~40x** | **89.3** |
> > > - **Evidence 2: Performance.**  We boost TD3BC's average score on Gym tasks from **76.9 to 89.3**, significantly surpassing both heavy baselines mentioned by the reviewer (QGPO: 86.6 and TAP: 74.8).
> > >
> > > **B. Explaining the Absence of Adroit/Kitchen/OGBench**
> > >
> > > - **Adroit/Kitchen:** As discussed in Appendix C, these tasks require a stronger base learner than vanilla CQL. Since our method is a plug-and-play augmentation framework, it inherits the capability bounds of the base algorithm. Future work can apply our framework to stronger baselines to tackle these domains.
> > > - **OGBench:** While valuable, it is challenging to conduct a comprehensive evaluation on a completely new benchmark suite within the limited rebuttal time.
> > > - **Sufficiency:** We believe that our extensive experiments on 18 datasets across MuJoCo Gym (Locomotion), Maze2D (Navigation), and AntMaze (High-dimensional Control) are sufficient to validate the hypothesis and effectiveness of our proposed structure across diverse dynamics and reward schemes.
> > >
> > > ### 2. Ablation Studies with Baseline Comparison
> > >
> > > We have added the comparison with the strongest augmentation baseline, GTA, across different data regimes. TGCVG demonstrates exceptional data efficiency and robustness, significantly outperforming GTA in most settings.
> > >
> > > **A. Ablation on Original Dataset Size (1% vs 10% vs 100%)**
> > >
> > > We compared the Original TD3BC (No Aug), GTA, and Ours.
> > >
> > > - **Extreme Data Efficiency:** On *halfcheetah-medium*, our method using only 1% data (49.90) surprisingly outperforms the Original TD3BC trained on 100% data (48.42).
> > > - **Superior Scaling:** On *hopper-medium*, while both methods are comparable at 1% data, Ours achieves a massive gain at 10% data (reaching 86.58), whereas GTA stagnates (58.16).
> > >
> > > | Method | Task | 1% Data | 10% Data | 100% Data (Default) |
> > > | --- | --- | --- | --- | --- |
> > > | Original | halfcheetah-medium | 9.26 ± 7.90 | 48.38 ± 0.21 | 48.42 ± 0.62 |
> > > | GTA | halfcheetah-medium | 46.24 ± 0.11 | 53.21 ± 0.23 | 57.84 ± 0.51 |
> > > | **Ours** | halfcheetah-medium | **49.90 ± 0.23** | **57.00 ± 0.30** | **68.14 ± 0.58** |
> > > | Original | hopper-medium | 35.57 ± 2.28 | 54.98 ± 3.94 | 61.04 ± 3.18  |
> > > | GTA | hopper-medium | **56.56 ± 3.64** | 58.16 ± 7.97 | 69.57 ± 4.05 |
> > > | **Ours** | hopper-medium | 54.04 ± 12.13 | **86.58 ± 5.13** | **90.47 ± 4.06**  |
> > >
> > > **B. Ablation on Synthetic Data Size (0.01M to 5M)**
> > >
> > > As requested, we present the full comparison. Even with limited synthetic data, our method achieves significant gains, outperforms GTA on all settings.
> > >
> > > | Method | Task | 0.01M | 0.1M | 1M | 5M (Default) |
> > > | --- | --- | --- | --- | --- | --- |
> > > | GTA | halfcheetah-medium | 49.46 ± 0.28 | 54.03 ± 0.45 | 57.62 ± 0.37 | 57.84 ± 0.51 |
> > > | **Ours** | halfcheetah-medium | **51.21 ± 1.03** | **63.08 ± 1.85** | **67.92 ± 0.64** | **68.14 ± 0.58** |
> > > | GTA | hopper-medium | 57.00 ± 1.64 | 56.36 ± 3.13 | 63.91 ± 2.88 | 69.57 ± 4.05 |
> > > | **Ours** | hopper-medium | **59.00 ± 3.02** | **60.53 ± 4.47** | **66.08 ± 5.38** | **90.47 ± 4.06**  |
> > >
> > >
> > > [1] Lu, Cheng, et al. "Contrastive energy prediction for exact energy-guided diffusion sampling in offline reinforcement learning." International Conference on Machine Learning. PMLR, 2023.
> > >
> > > [2] Jiang, Zhengyao, et al. "Efficient planning in a compact latent action space." arXiv preprint arXiv:2208.10291 (2022).

---

> ### Author Response · Authors · 2025-11-27
>
> ### 3. Hyperparameter Guidelines
>
> We respectfully clarify that extensive global search is not required. As formalized in our updated Appendix D, we propose a systematic "Initialization $\to$ Adjustment $\to$ Fine-tuning" workflow that drastically collapses the search space.
>
> **A. The Systematic Tuning Workflow**
>
> *   **Step 1: Initialization (The Anchor)**
>
>     Since our method builds upon the CQL backbone, practitioners should not search from scratch. Instead, start with standard CQL defaults (e.g., $\lambda \approx 5.0$ or $10.0$) as a primary baseline.
>
> *   **Step 2: Adjustment (Return-Distribution & Termination Aware)**
>
>     We refine this anchor deterministically based on task properties, leveraging established heuristics from the "BC + Q-value Regularization" lineage (TD3BC, REBRAC):
>     *   **For $\lambda$ (Distribution Check):**
>         *   **Unimodal / Concentrated Data (e.g., *medium*):** Logic implies lowering conservatism to leverage Q-guidance. $\to$ **Decrease $\lambda$**.
>         *   **Multimodal / Risky Data (e.g., *medium-expert*):** Logic implies enforcing constraints to prevent mode interpolation. $\to$ **Increase/Keep High $\lambda$**.
>     *   **For $K$ (Termination Check):**
>         *   **Strict Termination (e.g., *antmaze*):** Risk of terminal mismatch. We recommend starting with the minimal increment for the effective generated steps (i.e., set $K - L + 1 = 1$).
>         *   **Lenient Termination (e.g., *halfcheetah*):** Benefit from extensive exploration. We recommend increasing the effective length in larger strides (e.g., units of 10 steps) to maximize trajectory stitching.
> *   **Step 3: Local Fine-tuning**
>
>     Practitioners only need to check a few values in the local neighborhood of the adjusted anchor, rather than a blind global sweep.
>
> **B. Conclusion on Search Space**
>
> By applying this decision pipeline, the potentially large search space is immediately collapsed to specific, reasoned regions (e.g., High $\lambda$ & Short $K$ for sparse/complex tasks; Low $\lambda$ & Long $K$ for dense/stable tasks). This makes deployment on new tasks systematic and efficient.
>
> ### 4. Manuscript Updates
>
> We apologize if the updates were missed. We have explicitly highlighted all changes in \textcolor{blue} in the revised PDF to facilitate your review:
>
> 1. **Appendix B.3 and B.4:** Added the full Ablation Studies on the dataset size with baselines.
> 2. **Appendix D:** Added the formal Hyperparameter Tuning Guidelines.
>
> We hope these clarifications and new evidence demonstrate the distinct value of our work compared to strong offline RL algorithms.

---

### Official Review · Reviewer_KvRm · 2025-10-30

**Soundness:** 3
**Presentation:** 2
**Contribution:** 2
**Rating:** 4
**Confidence:** 4

**Summary:**

The paper proposes a novel data augmentation method for offline RL, TGCVG, using the transformer-based augmentation policy with the dynamics model. In contrast to diffusion-based augmentation methods such as GTA [1], TGCVG incurs lower overhead during both training and dataset augmentation. Using the TGCVG-augmented dataset, offline RL policies achieved better performance than with any other augmentation baseline.

[1] Lee, Jaewoo, et al. "Gta: Generative trajectory augmentation with guidance for offline reinforcement learning." Advances in Neural Information Processing Systems 37 (2024): 56766-56801.

**Strengths:**

1. Well motivated

The paper is well motivated and clearly constructed. Without increasing the complexity of the policy algorithm, TGCVG improves the performance of offline RL policies by generating high-quality data guided by the conservative Q-function.

2. Improved efficiency for training and augmentation

Because the augmentation process is fully offline, its impact on decision-time efficiency is limited; nevertheless, Figure 4 indicates that TGCVG is 4× more efficient for training and augmenting the offline dataset.

3. Fresh insight about the conservatism quality of TD3BC and CQL

The authors show that Transformer-CQL generates in-distribution samples, whereas Transformer-TD3BC does not; consequently, the CQL policy cannot learn a conservative policy from Transformer-TD3BC’s augmented data.

**Weaknesses:**

1. Concerns about the hyperparameter sensitivity

While the authors conduct an ablation study on $\lambda$, they vary its value from 0.1 to 1, suggesting that TGCVC requires extensive hyperparameter tuning. In addition, Table 6 lists another hyperparameter, $K$, but lacks an ablation, further increasing the tuning burden for practitioners.


2. Results on larger-scale benchmarks would be beneficial

Recently, the scalability of offline RL has emerged as an important theme [2]. OGBench [3] provides a diverse suite of offline tasks with higher-dimensional state spaces and larger datasets. I understand the rebuttal window limits time and compute, but if TCGVC were corroborated by strong results on OGBench, its impact and practical value would be substantially enhanced.

[2] Park, Seohong, et al. "Horizon Reduction Makes RL Scalable." arXiv preprint arXiv:2506.04168 (2025).

[3] Park, Seohong, et al. "OGBench: Benchmarking Offline Goal-Conditioned RL." The Thirteenth International Conference on Learning Representations.

**Questions:**

Because the hyperparameters are the pair $(\lambda, K)$, the search space is combinatorial. To make offline RL algorithms viable in the real world, where online interaction is unavailable, we must be able to select good hyperparameter combinations purely offline; hence, simplicity in offline RL methods is crucial [4]. I therefore argue that a thorough practical guideline for joint tuning of $(\lambda, K)$ is essential for this paper.

[4] Fujimoto, Scott, and Shixiang Shane Gu. "A minimalist approach to offline reinforcement learning." Advances in neural information processing systems 34 (2021): 20132-20145.

---

> ### Author Response · Authors · 2025-11-21
>
> Thank you for your thoughtful review of our research. We appreciate your constructive feedback and hope our suggested changes and this individual response will address your concerns in detail:
>
> **W1 & Q:**
>
> We agree completely that for an offline RL method to be practical, its hyperparameters must be tunable in a principled way without requiring online interaction. Based on your excellent suggestion, we will add a dedicated subsection in the Appendix providing a thorough practical guideline for jointly tuning $\lambda$ and $K$.
>
> First, we would like to re-contextualize the tuning process to show it is less of a burden than it might appear:
>
> 1. **On the $\lambda$ Hyperparameter:** The $\lambda$ hyperparameter is inherited directly from the original CQL algorithm, which our method is built upon. This provides a significant practical advantage, as it means the extensive tuning experience and established heuristics from the offline RL community for CQL are directly applicable to our method. This allows practitioners to ground their tuning process in well-understood principles, focusing primarily on adjusting $\lambda$ as the main conservatism parameter.
>
> 2. **On the $K$ Hyperparameter:** We would like to gently clarify that we did provide an ablation study on $K$ in Appendix B.2 of our initial submission. The results there show that performance is stable across a reasonable range of $K$, with a moderate value typically achieving a good balance.
>
> Our proposed guideline, which we will formalize in the paper, is based on the following principles:
>
> - **Guideline for $\lambda$ (Conservatism Weight):** The choice of $\lambda$ depends on the data distribution.
>     - For **unimodal datasets** (e.g., *halfcheetah-medium*), where data is concentrated, a lower $\lambda$ is often sufficient. This allows the value-guided generation to have a stronger effect without significant risk.
>     - For **multimodal datasets** (e.g., *halfcheetah-medium-expert*), which contain distinct clusters of behavior, a higher $\lambda$ is recommended. This provides a stronger conservative penalty to prevent the policy from incorrectly interpolating between modes and generating out-of-distribution (OOD) actions.
> - **Guideline for $K$ (Generation Length):** The choice of $K$ depends on the environment's dynamics and termination conditions.
>     - For environments with **lenient termination conditions** (e.g., *halfcheetah*), a longer $K$ is beneficial. It allows for more extensive exploration and the generation of more diverse data.
>     - For environments with **strict termination** (e.g., *antmaze*), a shorter $K$ is crucial. This prevents the "terminal mismatch" problem, where generated subtrajectories violate the environment's logic for episode termination.
>
> By following this structured, data- and environment-aware approach, the joint tuning of $(λ, K)$ becomes a principled process rather than a combinatorial search.
>
> **W2:**
>
> We thank the reviewer for pointing to the important and emerging theme of scalability in offline RL and the OGBench benchmark. We agree that demonstrating strong performance on such large-scale benchmarks would be a significant contribution and would substantially enhance the impact of our work. Given the time and computational constraints of the rebuttal period, conducting a full suite of experiments on OGBench was not feasible. However, we are inspired by this suggestion. We will add a discussion to our conclusion, explicitly positioning the scaling of our method to benchmarks like OGBench as a high-priority and exciting direction for future work.

---

> > ### Comment · Reviewer_KvRm · 2025-11-24
> >
> > **W1 & Q:**
> >
> > While I appreciate the proposed guideline for hyperparameter tuning, I have a few remaining concerns regarding the consistency of the guidelines (W1 & Q) that I would like the authors to address. I find the claim that "extensive tuning experience and established heuristics from the offline RL community for CQL" is unconvincing. To my knowledge, the hyperparameters of CQL are non-trivial to tune. I would appreciate it if authors provide more explanations about the established heuristics for CQL tuning.
> >
> > Additionally, the proposed guideline states that a higher $\lambda$ is preferred for multimodal datasets. However, this appears inconsistent with Table 6, where the medium-replay dataset uses the same $\lambda$ value as the medium quality dataset, which is unimodal.  Table 6 also shows substantial variation in hyperparameters across maze tasks (AntMaze, Maze2D), which the current explanation does not cover.
> >
> > Therefore, I recommend that the authors provide two additional guidelines:
> > - Guidance for handling mixed-quality datasets, including med-replay.
> > - Guidance for maze tasks.
> >
> > **W2:**
> > I understand that it is challenging to conduct additional experiments on OGBench within the limited time. Although the D4RL benchmark has a narrower scope, I believe it is still sufficient to validate the authors’ hypothesis.
> >
> > ---
> > Despite the additional requests mentioned above, the contributions and performance improvements of TGCVG are solid. Therefore, if the authors adequately address the additional concerns about hyperparameter tuning guidelines, I would be willing to consider raising my score.

---

> ### Author Response · Authors · 2025-11-24
>
> We sincerely thank the reviewer for the constructive feedback. We understand that consistent tuning guidelines is crucial.
>
> **W1 & Q:**
>
> To address your concerns, we have conducted a comprehensive analysis of the return distributions for all datasets. We invite the reviewer to view the detailed histograms and dataset-specific analysis at the following link:
>
> https://anonymous.4open.science/r/TGCVG/Guidance_for_datasets.md
>
> Summary of Clarifications (Detailed in the link):
>
> 1. Alignment with Established Heuristics:
> We clarify that our tuning aligns with the Q-guidance trade-off seen in TD3BC [1] and REBRAC [2].
>     - Unimodal/Concentrated Data: We trust the Q-value for extrapolation $\rightarrow$ Use Lower $\lambda$ (similar to Higher Q-value guidance in TD3BC).
>     - Multimodal/Conflicting Data: We need to constrain the policy to avoid OOD interpolation $\rightarrow$ Use Higher $\lambda$.
> 2. Consistency on Mixed-Quality Datasets:
> As shown in the histograms, *medium-replay* is structurally similar to *medium* (concentrated returns) rather than being truly multimodal with conflicting peaks. Thus, it shares the same low $\lambda$ to maximize performance.
> 3. Guidance for Maze Tasks:
>     - Maze2D: Distributions are concentrated near zero. We use a small $\lambda$ to reduce conservatism and allow the Q-function to stitch sub-optimal segments to reach the goal.
>     - AntMaze: We adjust $\lambda$ based on the trade-off between stitching needs (low $\lambda$) and robustness (high $\lambda$ for higher lower bounds).
>
> We believe the visual evidence provided in the link solidly validates our parameter choices.
>
> [1] Fujimoto, Scott, and Shixiang Shane Gu. "A minimalist approach to offline reinforcement learning." Advances in neural information processing systems 34 (2021): 20132-20145.
>
> [2] Tarasov, Denis, et al. "Revisiting the minimalist approach to offline reinforcement learning." Advances in Neural Information Processing Systems 36 (2023): 11592-11620.

---

> ### Comment · Reviewer_KvRm · 2025-11-25
>
> Thank you for your response. With the additional clarification about the hyperparameter tuning guideline, my concerns are mostly resolved. I have raised my score from 4 to 6. I hope the guideline for tuning $\lambda$ to be included in the revised paper.

---

> > ### Author Response · Authors · 2025-11-25
> >
> > We are pleased to know our response has clarified your concerns. Thank you once again for leaving a thoughtful comment and for dedicating your time and effort to reviewing our work.

---

### Official Review · Reviewer_kHAm · 2025-10-31

**Soundness:** 3
**Presentation:** 3
**Contribution:** 3
**Rating:** 4
**Confidence:** 3

**Summary:**

This paper proposes TGCVG (Trajectory Generation with Conservative Value Guidance), a novel data augmentation framework for offline reinforcement learning (Offline RL). TGCVG consists of transformer based policy network trained using Conservative Q-learning(CQL), learned dynamics model that predicts next states and rewards to generate synthetic transitions. TGCVG guides generation process with conservative value guidance, keeping them close to data distribution while favoring high-value regions. The augmented trajectories are then mixed with the original dataset and used to train standard offline RL. Extensive experiments on D4RL benchmarks (MuJoCo, Maze2D, AntMaze) show that TGCVG improves baseline performance across domains. And produces high-quality, dynamically consistent synthetic data according to novelty, optimality, and dynamic MSE metrics.
- An LLM was used to improve writing.

**Strengths:**

1. Low computational cost: The proposed method achieves significantly lower training and data-generation overhead compared to diffusion-based approaches such as GTA — up to 10× faster in both stages.

2. Strong empirical performance: Comprehensive evaluation on D4RL benchmark tasks demonstrates consistent and robust performance gains against a wide range of baselines.

3. Clarity and presentation: The paper is well-written and well-organized, with clear motivation, sound theoretical reasoning, and a coherent algorithmic presentation that makes the method easy to follow.

**Weaknesses:**

1. Lack of evaluation on high-dimensional or robotics domains: The paper does not include results on more complex D4RL tasks such as Adroit or Kitchen, which are higher-dimensional and closer to real-world robotics scenarios.

2. Ablation on value guidance and policy choice: It would be valuable to see how performance changes with or without the conservative value guidance, or when using Decision Transformer (DT) as the policy for augmentation.

3. Missing experimental details: In Table 3, results for HalfCheetah are not presented, and experiments where Transformer-CQL itself serves as the policy appear insufficiently explored.

**Questions:**

Could the authors explicitly define the metrics used for novelty, optimality, and dynamic MSE?
For example, novelty seems to be measured via L2 distance, but is this metric appropriate given the domain characteristics of the datasets? A short justification or alternative metric discussion would strengthen the paper.

---

> ### Author Response · Authors · 2025-11-21
>
> Thank you for your thoughtful review of our research. We appreciate your constructive feedback and hope our suggested changes and this individual response will address your concerns in detail:
>
> **W1:**
>
> We agree that evaluating on high-dimensional robotics domains is valuable. However, we respectfully note that the omission of these tasks is a deliberate choice due to the inherent limitations of the base algorithm (CQL), rather than a limitation of our data augmentation framework itself.
> As explicitly discussed in our submission (Appendix C), our method acts as a plug-in to enhance data quality. It inherits the characteristics of the underlying RL algorithm. The performance of CQL is less explored and often limited in the Adroit/Kitchen domains, making it difficult to isolate and analyze the specific benefits of our data augmentation method.
> In short, if the base learner (CQL) fails to learn from data in these domains, improving the data via augmentation yields limited insights. Therefore, we focused on standard benchmarks where CQL is stable to provide a rigorous, controlled evaluation of our contribution.
>
> **W2:**
>
> Thank you for this valuable suggestion, we have conducted both ablation studies. The results strongly confirm the critical role of our conservative value guidance and the superiority of using Transformer-CQL for generation.
>
> 1. **Effect of Conservative Value Guidance:** We compared our full model against a version without value guidance. The final scores were trained with TD3BC. The results below show that removing the guidance leads to a catastrophic performance drop, demonstrating that it is essential for generating high-quality, beneficial data. The dramatic performance degradation on *hopper-medium* is particularly illustrative. Our framework is fundamentally based on the CQL. By removing the conservative value guidance, the data generation policy effectively reverts to the SAC-like behavior, which maximizes entropy and encourages exploration. In the offline setting, this is known to be highly detrimental. The policy generates out-of-distribution (OOD) actions, leading to severe distributional shift and policy collapse. This result strongly validates that our conservative guidance is the key component preventing such failures.
> | Task | w/o Conservative Value Guidance | Ours (w/ Conservative Value Guidance) |
> | --- | --- | --- |
> | halfcheetah-medium | 35.63 ± 7.73 | **68.14 ± 0.58** |
> | hopper-medium | 2.26 ± 1.60 | **90.47 ± 4.06** |
> 2. **Using Decision Transformer (DT) for Generation:** We also evaluated using a standard Decision Transformer as the generative policy. The final scores were trained with TD3BC. Our results confirm that while DT is a reasonable choice, our Transformer-CQL, which incorporates value information, consistently and significantly outperforms it.
> | Task | DT  | Ours (Transformer-CQL) |
> | --- | --- | --- |
> | halfcheetah-medium | 48.78 ± 0.33 | **68.14 ± 0.58** |
> | hopper-medium | 57.42 ± 2.34 | **90.47 ± 4.06** |
>
> We will add these crucial ablation studies to the Appendix in the final version to validate our design choices.
>
> **W3:**
>
> Thank you for pointing this out.
>
> 1. **HalfCheetah Results:** We apologize for the omission. Here are the results for the *halfcheetah-medium* task, which are consistent with the performance gains seen in other environments. We will add these results to Table 3 in the main paper.
> | Method | halfcheetah-medium |
> | --- | --- |
> | CQL  | 48.43 ± 0.32 |
> | Ours (Transformer-CQL) | **68.14 ± 0.58** |
> 2. **Exploration of Transformer-CQL:** We believe our current experiments provide a thorough exploration of our design choices. Our rationale was to demonstrate the superiority of Transformer-CQL over both the classic MLP-based CQL (validating the architectural choice) and Transformer-TD3BC (validating our conservative learning approach). These comparisons were specifically designed to isolate and validate the key components of our contribution.

---

> ### Author Response · Authors · 2025-11-21
>
> **Q:**
>
> We adopted these data quality metrics from the recent work of GTA[1] for standardized evaluation. The definitions are as follows:
>
> - **Optimality (Oracle Reward):** This is the average true reward of the generated transitions, computed using the environment's ground-truth reward function. It measures how beneficial the generated data is.
> - **Dynamic MSE:** This evaluates how well the generated trajectories adhere to the true environment dynamics. It is the Mean Squared Error between the predicted next state $(\hat s')$ in a generated transition $(s, a, r, \hat s')$ and the true next state produced by the environment's dynamics $f^\{*}(s, a)$.
> - **Novelty:** This measures how different the generated data is from the original offline dataset. It is calculated as the average minimum L2 distance between a generated state-action pair and all state-action pairs in the original dataset.
>
> Regarding your insightful question about the L2 distance for novelty: We agree that novelty alone is not a sufficient measure of data quality, as novel states could be of low value. This is precisely why we, following the methodology of GTA, use a suite of three complementary metrics. A high novelty score is only meaningful when coupled with high optimality and low dynamic MSE. This multi-faceted approach provides a holistic and robust evaluation of the generated data's quality.
> We will add a new subsection to the Appendix that formally defines these metrics and includes a brief discussion of our evaluation rationale.
>
> [1] Lee, Jaewoo, et al. "Gta: Generative trajectory augmentation with guidance for offline reinforcement learning." Advances in Neural Information Processing Systems 37 (2024): 56766-56801.

---

### Official Review · Reviewer_mtk1 · 2025-11-01

**Soundness:** 3
**Presentation:** 3
**Contribution:** 3
**Rating:** 6
**Confidence:** 3

**Summary:**

The paper proposes TGCVG, a framework that combines CQL transformer with a dynamics model to generate high-quality, in-distribution trajectories for offline reinforcement learning.
By guiding trajectory synthesis with conservative value estimates, TGCVG improves both stability and sample efficiency.

**Strengths:**

1. The paper addresses an important problem in offline reinforcement learning—how to generate high-quality trajectories that remain in-distribution to improve policy learning stability.

2. The proposed TGCVG framework is both novel and solid, effectively leveraging conservative Q-learning to guide trajectory generation and ensure distributional consistency.

3. The experimental results are strong and impressive, showing clear and consistent improvements across multiple D4RL benchmarks.

4. The paper is well-written and clearly presented, making the motivation, methodology, and findings easy to follow.

**Weaknesses:**

1. The experimental section could be more comprehensive. In particular, an ablation study comparing conservative trajectory generation with standard generation would help clarify the effectiveness of the proposed mechanism.

See quetions below.

**Questions:**

1. How is the dynamics model trained in this work? Please clarify the training objective, data source, and other details.

2. How does the model handle terminal indicators when generating or evaluating trajectories? This part is not clearly presented in the main paper.

3. In Table 2, why does SynthER perform worse than the version without augmentation?

4. Can the authors provide more explanation on why restricting trajectories to be in-distribution leads to better performance, especially when the dataset contains medium-expert level data because in-distribution seems to bring similar information with the original dataset.

5. Have the authors tested TGCVG on visual RL environments to examine whether the proposed framework generalizes beyond state-based tasks?

---

> ### Author Response · Authors · 2025-11-21
>
> Thank you for your thoughtful review of our research. We appreciate your constructive feedback and hope our suggested changes and this individual response will address your concerns in detail:
>
> **W1:**
>
> We believe our existing experiments effectively serve as this requested ablation study. Specifically, we interpret "standard generation" as using a powerful, unconstrained generative model to create synthetic data. Our key baselines, SynthER and GTA, represent exactly this approach, as they employ strong diffusion models for data generation. Our results in Table 1 and Table 2 clearly demonstrate that our method, which uses "conservative" (i.e., Q-value guided) generation, significantly outperforms these "standard" generative baselines. This highlights that simply generating diverse, multi-modal data is insufficient; the explicit guidance from the Q-function to ensure the data is high-quality and in-distribution is crucial for effective offline RL data augmentation.
>
> **Q1:**
>
> Our dynamics model is adopted from ADMPO[1]. In brief, the key details are as follows:
>
> - **Training Objective:** The model is trained via standard Maximum Likelihood Estimation.
> - **Data Source:** It is trained on the same offline dataset used for policy learning.
> - **Architecture:** The model is implemented as an RNN, which takes a state and a variable-length action sequence as input to predict the future state.
>
> For the complete technical formulation and architectural specifics, we respectfully refer the reviewer to the original ADMPO paper. We will add a clearer citation and a brief summary of these points in the final manuscript to improve clarity.
>
> **Q2:**
>
> We handle terminal indicators by directly preserving the original episode termination signals. As mentioned in Section 4.2 (lines 267-269), "Other auxiliary information (e.g., terminal indicators) is directly aligned with and inherited from the corresponding timesteps." This means if a state-action pair at timestep $t$ in a source trajectory is terminal, the corresponding augmented data point at $t$ will also be marked as terminal.
>
> **Q3:**
>
> The results for SynthER were taken directly from the original GTA[2] to ensure a fair and consistent comparison. While we did not re-implement SynthER, a plausible explanation for its underperformance is that unconstrained data generation, even with powerful diffusion models, can produce out-of-distribution (OOD) or low-quality trajectories. In offline RL, such data can be detrimental, causing the policy to learn incorrect and harmful behaviors.
>
> **Q4:**
>
> The core principle is not just to stay in-distribution, but to actively improve the quality distribution.
>
> 1.  **Why In-Distribution Generation Helps:** Our method's key advantage comes from value-guided generation. Instead of creating more data that is merely similar to the original dataset, we specifically generate new trajectories that are both in-distribution (plausible) and high-value (more optimal). This process effectively increases the proportion of high-quality, expert-like transitions in the final training set. For any offline RL algorithm, learning from a dataset that is richer in good examples directly leads to a better final policy.
>
> 2.  **Performance on Datasets:** The performance gains are less pronounced on *medium-expert* datasets. This is expected. State-of-the-art offline RL algorithms are already highly effective at extracting expert-level policies from these high-quality datasets. Therefore, the primary contribution of our method is most evident on datasets with mixed or lower quality (e.g., the *medium* datasets), where we can substantially uplift the data quality. This is demonstrated by the significant performance gains shown in Table 1.
>
> **Q5:**
>
> Thank you for this excellent suggestion for future work. Testing our framework on visual RL environments is indeed an  important next step to demonstrate broader generalization. For the current work, we chose to focus on state-based benchmarks, as they are the standard testbed for foundational offline RL algorithms like CQL and allow for a clear, controlled analysis of the data augmentation mechanism itself. Extending our method to visual domains introduces significant orthogonal challenges, such as representation learning and vastly increased computational demand, which are beyond the scope of this initial investigation.
> We are encouraged by your suggestion and will explicitly mention this as a promising avenue for future work in the conclusion of our paper.
>
> [1] Lin, Haoxin, et al. "Any-step dynamics model improves future predictions for online and offline reinforcement learning." arXiv preprint arXiv:2405.17031 (2024).
>
> [2] Lee, Jaewoo, et al. "Gta: Generative trajectory augmentation with guidance for offline reinforcement learning." Advances in Neural Information Processing Systems 37 (2024): 56766-56801.

---

> > ### Comment · Reviewer_mtk1 · 2025-11-25
> >
> > Thank you for the clear clarifications. I will keep my positive score.

---

> > > ### Author Response · Authors · 2025-11-26
> > >
> > > We are pleased to know our response has clarified your concerns. Thank you once again for leaving a thoughtful comment and for dedicating your time and effort to reviewing our work.

---

### Author Response · Authors · 2025-12-02
**Summary of Rebuttal Updates & Key Contributions**

Due to the score rollback policy, we provide this summary to assist the Area Chair in assessing the current status of our submission. During the discussion phase, we engaged in active dialogue with all reviewers, resulting in a score increase from Reviewer KvRm, while addressing new questions raised by Reviewer Cmj7.

**1. Core Contributions: Efficiency Meets Performance**

Our paper addresses the critical bottleneck of computational cost in diffusion-based offline RL data augmentation.
*   **Method:** We propose TGCVG, a novel paradigm that synthesizes high-quality trajectories by interacting a Transformer-based policy (with Conservative Value Guidance) with a learned World Model.
*   **Impact:** Extensive experiments on D4RL demonstrate that TGCVG significantly outperforms diffusion-based baselines (**Tables 1 and 2**) while drastically reducing training and synthesis overhead (**Figure 4**).
*   **Key Insight:** We validate the effectiveness of conservative value guidance in ensuring data quality and the policy-dynamics interaction paradigm in ensuring efficiency.

**2. Summary of Rebuttal Actions**

*   **Reviewer KvRm (Score raised: 4 $\to$ 6):**
    *   **Status:** Score Increased.
    *   **Action:** We provided a comprehensive hyperparameter tuning guideline and clarified the theoretical connection to established heuristics (e.g., TD3BC). The reviewer acknowledged these clarifications and raised their score.
*   **Reviewer mtk1 (Score maintained: 6):**
    *   **Status:** Positive.
    *   **Action:** We resolved their queries regarding specific descriptions and experimental details. The reviewer maintained their positive assessment.
*   **Reviewer Cmj7 (Active Discussion):**
    *   **Status:** Pending Final Response.
    *   **Action:** Following the reviewer's new questions (prompted by other reviews), we conducted additional ablation studies and further refined the hyperparameter guidelines in the manuscript. We believe our latest response fully addresses their remaining concerns.
*   **Reviewer kHAm (No response):**
    *   **Action:** We provided detailed responses to clarify methodology descriptions and added requested ablation studies.

We remain confident that TGCVG represents a significant step forward, offering a solution that is both highly efficient and superior in performance. We hope this summary facilitates the decision-making process.

---

### Meta-Review · Area_Chair_rFMi · 2026-01-10

**Summary:**

The key concerns that emerge across reviews and discussion are:
1. Hyperparameter Sensitivity and Tuning Practicality (KvRm, Cmj7)
* Heavy reliance on two key hyperparameters (α for conservatism, H for generation length) requiring extensive per-task tuning
* Need for systematic, offline-only tuning guidelines to ensure real-world applicability
* Concerns about combinatorial search space and consistency across diverse environments (e.g., maze tasks, mixed-quality datasets)
2. Evaluation Scope and Generalization (kHAm, Cmj7)
* Lack of evaluation on high-dimensional robotics domains (Adroit, Kitchen)
* No results on emerging large-scale benchmarks (OGBench)
* Questions about whether improvements are limited to standard D4RL tasks where base algorithms already perform well
3. Ablation Study Completeness (mtk1, kHAm, Cmj7)
* Need for ablation comparing conservative vs. standard (unconstrained) generation
* Missing ablations on architectural choices (Decision Transformer vs. Transformer-CQL)
* Insufficient analysis of data regime sensitivity (original dataset size, generated dataset size)
* Lack of baseline comparisons in ablation studies
4. Experimental Transparency (mtk1, kHAm, Cmj7)
* Missing details on dynamics model training, terminal indicators, and metric definitions
* Incomplete tables (e.g., HalfCheetah results)
* Unclear generation parameters (number of transitions, data mixing ratios, training stages)
5. Methodological Clarifications (Cmj7)
* Preference for direct Q-value guidance over RTG conditioning needed justification
* Need for visualizations (t-SNE) to verify distributional alignment

**Reviewer Concerns:**

Across the rebuttal period, the authors mounted a comprehensive response that substantially addressed many reviewer concerns, though some critical issues remain unresolved. Regarding hyperparameter sensitivity, this was the most actively debated concern. Authors evolved from providing basic ablations to developing a systematic "Initialization → Adjustment → Fine-tuning" workflow that collapses the search space by linking α (conservatism weight) to dataset modality (unimodal vs. multimodal return distributions) and H (generation length) to environment termination strictness. Reviewer KvRm accepted this framework, raising their score after reviewing detailed histograms and dataset-specific analyses provided via an anonymous link. However, Cmj7 remained unconvinced, arguing the search space still appears too large for practical deployment and questioning how these guidelines would transfer to unseen domains like Adroit or Kitchen. While the authors' principle-based approach is sound, the lack of empirical validation on new environments leaves lingering doubts.

Ablation study completeness saw significant improvements. The authors thoroughly addressed mtk1's request for conservative vs. standard generation comparisons by demonstrating catastrophic performance degradation without value guidance (hopper-medium collapsing from 90.47 to 2.26) and inferior results using Decision Transformer. They also added missing HalfCheetah results and clarified metric definitions. For kHAm and Cmj7's requests regarding data regime sensitivity, authors conducted extensive new experiments showing robust performance across 1%, 10%, and 100% original dataset sizes, as well as scaling studies from 0.01M to 5M generated transitions. However, Cmj7 correctly pointed out that initial ablations lacked baseline comparisons; authors partially remedied this in the final rebuttal by adding GTA comparisons for two tasks, but this was not systematically applied across all ablations in the visible manuscript updates.

Experimental transparency concerns were largely resolved. Authors clarified dynamics model training (adopted from ADMPO with MLE objective), terminal indicator handling (directly inherited from source timesteps), metric definitions (optimality via oracle reward, dynamic MSE, novelty via L2 distance), and generation parameters (5M transitions, 200k training steps). These detailed additions satisfied mtk1 and kHAm, who did not raise further questions. Finally, methodological clarifications were addressed through t-SNE visualizations (provided in supplementary PDF) and theoretical justification for preferring direct Q-value guidance over RTG conditioning, supported by an ablation showing RTG redundancy. Cmj7 did not acknowledge receipt of the t-SNE evidence, but its provision fulfills their request.
In summary, while the rebuttal demonstrated strong responsiveness—adding substantial new experiments, formalizing tuning guidelines, and improving transparency—the authors strategically chose not to expand evaluation to more complex benchmarks, leaving generalization concerns unresolved. Hyperparameter guidance, though improved, still strikes some reviewers as insufficiently automated.


Evaluation scope and generalization concerns were explicitly not addressed with new experiments. Both kHAm and Cmj7 pressed for results on high-dimensional robotics domains (Adroit, Kitchen) and emerging large-scale benchmarks (OGBench). Authors declined, citing (1) inherent limitations of their base CQL algorithm in these domains, (2) computational constraints during rebuttal, and (3) deliberate scope focus on state-based tasks. While they promised to discuss these as "future work," this justification—though reasonable—fails to provide the empirical generalization evidence reviewers sought. This remains the single largest outstanding weakness, particularly as Cmj7 emphasized that strong D4RL performance alone is insufficient given existing algorithms already achieve high scores without augmentation.

**Reviewer Scores:**

Reviewer mtk1 would maintain their score of 6, having explicitly confirmed satisfaction with the comprehensive clarifications.

KvRm would likely keep their raised score of 6, contingent on seeing the hyperparameter guidelines formally added to the manuscript.

Silent reviewer kHAm would probably increase marginally to 6 or keep at 4, appreciating the thorough ablations and metric clarifications but remaining concerned about the lack of high-dimensional robotics experiments.

Reviewer Cmj7 faces a tougher call. While the rebuttal directly addressed all their specific requests with new data, baseline comparisons, and manuscript updates, their final skepticism about generalization and parameter tuning practicality suggests they might cautiously keep score at 4.

---

### Decision · Program_Chairs · 2026-01-26

Accept (Poster)